# Adaptivity of deep ReLU network for learning in Besov and mixed smooth Besov spaces: optimal rate and curse of dimensionality

**Taiji Suzuki**
The University of Tokyo, Tokyo, Japan
Center for Advanced Intelligence Project, RIKEN, Tokyo, Japan
Japan Digital Design, Tokyo, Japan
`taiji@mist.i.u-tokyo.ac.jp`

## Abstract

Deep learning has shown high performances in various types of tasks from visual recognition to natural language processing, which indicates superior flexibility and adaptivity of deep learning. To understand this phenomenon theoretically, we develop a new approximation and estimation error analysis of deep learning with the ReLU activation for functions in a Besov space and its variant with mixed smoothness. The Besov space is a considerably general function space including the Hölder space and Sobolev space, and especially can capture spatial inhomogeneity of smoothness. Throughout the analysis in the Besov space, it is shown that deep learning can achieve the minimax optimal rate and outperform any non-adaptive (linear) estimator such as kernel ridge regression, which shows that deep learning has higher adaptivity to the spatial inhomogeneity of the target function than other estimators such as linear ones. In addition to this, it is shown that deep learning can avoid the curse of dimensionality if the target function is in a *mixed smooth* Besov space. We also show that the dependency of the convergence rate on the dimensionality is tight due to its minimax optimality. These results support high adaptivity of deep learning and its superior ability as a feature extractor.

## 1 Introduction

Deep learning has shown great success in several applications such as computer vision and natural language processing. As its application range is getting wider, theoretical analysis to reveal the reason why deep learning works so well is also gathering much attention. To understand deep learning theoretically, several studies have been developed from several aspects such as approximation theory and statistical learning theory. A remarkable property of neural network is that it has universal approximation capability even if there is only one hidden layer (Cybenko, 1989; Hornik, 1991; Sonoda & Murata, 2017). Thanks to this property, deep and shallow neural networks can approximate any function with any precision (of course, the meaning of the terminology "any" must be rigorously defined like "any function in $L^1(\mathbb{R})$"). A natural question coming next is its expressive power. It is shown that the expressive power of deep neural network grows exponentially against the number of layers (Montufar et al., 2014; Bianchini & Scarselli, 2014; Cohen et al., 2016; Cohen & Shashua, 2016; Poole et al., 2016) where the "expressive power" is defined by several ways.

The expressive power of neural network can be analyzed more precisely by specifying the target function's property such as smoothness. Barron (1993; 1994) developed an approximation theory for functions having limited "capacity" that is measured by integrability of their Fourier transform. An interesting point of the analysis is that the approximation error is not affected by the dimensionality of the input. This observation matches the experimental observations that deep learning is quite effective also in high dimensional situations. Another typical approach is to analyze function spaces with smoothness conditions such as the Hölder space. In particular, deep neural network with the ReLU activation (Nair & Hinton, 2010; Glorot et al., 2011) has been extensively studied recently from the view point of its expressive power and its generalization error. For example, Yarotsky (2017) derived the approximation error of the deep network with the ReLU activation for

Table 1: Comparison between the performances of deep learning and linear methods. $N$ is the number of parameters to approximate a function in a Besov space ($B_{p,q}^s([0,1]^d)$), and $n$ is the sample size. The approximation error is measured by $L^r$-norm. The $\tilde{O}$ symbol hides the poly-log order.

| Model | Deep learning | Linear method |
|---|---|---|
| Approximation error rate | $\tilde{O}(N^{-\frac{s}{d}})$ | $\tilde{O}\left(N^{-\frac{s}{d}+(\frac{1}{p}-\frac{1}{r})_+}\right)$ |
| Estimation error rate | $\tilde{O}(n^{-\frac{2s}{2s+d}})$ | $\Omega\left(n^{-\frac{2s-(2/(p\wedge 1)-1)}{2s+1-(2/(p\wedge 1)-1)}}\right)$ |

functions in the Hölder space. Schmidt-Hieber (2018) evaluated the estimation error of regularized least squares estimator performed by deep ReLU network based on this approximation error analysis in a nonparametric regression setting. Petersen & Voigtlaender (2017) generalized the analysis of Yarotsky (2017) to the class of *piece-wise* smooth functions. Imaizumi & Fukumizu (2018) utilized this analysis to derive the estimation error to estimate the piece-wise smooth function and concluded that deep leaning can outperform linear estimators in that setting. Although these error analyses are standard from a nonparametric statistics view point and the derived rates are known to be (near) minimax optimal, the analysis is rather limited because they are given mainly based on the Hölder space. However, there are several other function spaces such as the Sobolev space and the space of finite total variations. A comprehensive analysis to deal with such function classes from a unified view point is required.

In this paper, we give generalization error bounds of deep ReLU networks for a *Besov space* and its variant with *mixed smoothness*, which includes the Hölder space, the Sobolev space, and the function class with total variation as special cases. By doing so, (i) we show that deep learning[1] achieves the minimax optimal rate on the Besov space and notably it outperforms *any linear estimator* such as the kernel ridge regression, and (ii) we show that deep learning can *avoid the curse of dimensionality* on the mixed smooth Besov space and achieves the minimax optimal rate. As related work, Mhaskar & Micchelli (1992); Mhaskar (1993); Chui et al. (1994); Mhaskar (1996); Pinkus (1999) also developed an approximation error analysis which essentially leads to analyses for Besov spaces. However, the ReLU activation is basically excluded and comprehensive analyses for the Besov space have not been given. As a summary, the contribution of this paper is listed as follows:

(i) To investigate adaptivity of deep learning, we give an explicit form of approximation and estimation error bounds for deep learning with the ReLU activation where the target functions are in the Besov spaces ($B_{p,q}^s$) for $s > 0$ and $0 < p, q \leq \infty$ with $s > d(1/p - 1/r)_+$ where $L^r$-norm is used for error evaluation. In particular, deep learning outperforms any linear estimator such as kernel ridge regression if the target function has highly spatial inhomogeneity of its smoothness. See Table 1 for the overview.

(ii) To investigate the effect of dimensionality, we analyze approximation and estimation problems in so-called the mixed smooth Besov space by ReLU neural network. It is shown that deep learning with the ReLU activation can ease the curse of dimensionality and achieve the near minimax optimal rate. The theory is developed on the basis of the *sparse grid* technique (Smolyak, 1963). See Table 2 for the overview.

## 2 Set up of function spaces

In this section, we define the function classes for which we develop error bounds. In particular, we define the Besov space and its variant with mixed smoothness. The typical settings in statistical learning theory is to estimate a function with a *smoothness* condition. There are several ways to characterize "smoothness." Here, we summarize the definitions of representative functional spaces that are appropriate to define the smoothness assumption.

Let $\Omega \subset \mathbb{R}^d$ be a domain of the functions. Throughout this paper, we employ $\Omega = [0,1]^d$. For a function $f : \Omega \to \mathbb{R}$, let $\|f\|_p := \|f\|_{L^p(\Omega)} := (\int_\Omega |f|^p \mathrm{d}x)^{1/p}$ for $0 < p < \infty$. For $p =$

---

[1]In this paper, we mainly consider a regularized empirical risk minimization given in Eq. (7) as an estimation procedure, and say "deep learning" to indicate it.

Table 2: Summary of relation between related existing work and our work for a mixed smooth Besov space. $N$ is the number of parameters in the deep neural network, $n$ is the sample size. $\beta$ represents the smoothness parameter, and $d$ represents the dimensionality of the input. The approximation accuracy is measured by $L^2$-norm and estimation accuracy is measured by the square of $L^2$-norm. See Theorem 3 for the definition of $u$.

| Function class | Hölder | Barron class | m-Sobolev $(0 < \beta \leq 2)$ | m-Besov $(0 < \beta)$ |
|---|---|---|---|---|
| Approximation | | | | |
| Author | Yarotsky (2017), Liang & Srikant (2016) | Barron (1993) | Montanelli & Du (2017) | This work |
| Approx. error | $\tilde{O}(N^{-\frac{\beta}{d}})$ | $\tilde{O}(N^{-1/2})$ | $\tilde{O}(N^{-\beta})$ | $\tilde{O}(N^{-\beta})$ |
| Estimation | | | | |
| Author | Schmidt-Hieber (2018) | Barron (1993) | —– | This work |
| Estimation error | $\tilde{O}(n^{-\frac{2\beta}{2\beta+d}})$ | $\tilde{O}(n^{-\frac{1}{2}})$ | —– | $\tilde{O}(n^{-\frac{2\beta}{2\beta+1}} \times \log(n)^{\frac{2(d-1)(u+\beta)}{1+2\beta}})$ |

$\infty$, we define $\|f\|_\infty := \|f\|_{L^\infty(\Omega)} := \sup_{x \in \Omega} |f(x)|$. For $\alpha \in \mathbb{R}^d$, let $|\alpha| = \sum_{j=1}^d |\alpha_j|$. Let $\mathcal{C}^0(\Omega)$ be the set of continuous functions equipped with $L^\infty$-norm: $\mathcal{C}^0(\Omega) := \{f : \Omega \to \mathbb{R} \mid f$ is continuous and $\|f\|_\infty < \infty\}$ [2]. For $\alpha \in \mathbb{Z}_+^d$, we denote by $D^\alpha f(x) = \frac{\partial^{|\alpha|} f}{\partial^{\alpha_1} x_1 \ldots \partial^{\alpha_d} x_d}(x)$ [3].

**Definition 1** (Hölder space ($\mathcal{C}^\beta(\Omega)$)). *Let $\beta > 0$ with $\beta \notin \mathbb{N}$ be the smoothness parameter. For an $m$ times differentiable function $f : \mathbb{R}^d \to \mathbb{R}$, let the norm of the Hölder space $\mathcal{C}^\beta(\Omega)$ be $\|f\|_{\mathcal{C}^\beta} := \max_{|\alpha| \leq m} \|D^\alpha f\|_\infty + \max_{|\alpha|=m} \sup_{x,y \in \Omega} \frac{|D^\alpha f(x) - D^\alpha f(y)|}{|x-y|^{\beta-m}}$, where $m = \lfloor \beta \rfloor$ (the largest integer less than $\beta$). Then, ($\beta$-)Hölder space $\mathcal{C}^\beta(\Omega)$ is defined as $\mathcal{C}^\beta(\Omega) = \{f \mid \|f\|_{\mathcal{C}^\beta} < \infty\}$.*

The parameter $\beta > 0$ controls the "smoothness" of the function. Along with the Hölder space, the *Sobolev space* is also important.

**Definition 2** (Sobolev space ($W_p^k(\Omega)$)). *Sobolev space ($W_p^k(\Omega)$) with a regularity parameter $k \in \mathbb{N}$ and a parameter $1 \leq p \leq \infty$ is a set of functions such that the Sobolev norm $\|f\|_{W_p^k} := (\sum_{|\alpha| \leq k} \|D^\alpha f\|_p^p)^{\frac{1}{p}}$ is finite (where the derivative $D$ is taken in the weak sense).*

There are some ways to define a Sobolev space with fractional order, one of which will be defined by using the notion of *interpolation space* (DeVore, 1998; Adams & Fournier, 2003), but we don't pursue this direction here. Finally, we introduce *Besov space* which further generalizes the definition of the Sobolev space. To define the Besov space, we introduce the modulus of smoothness.

**Definition 3.** *For a function $f \in L^p(\Omega)$ for some $p \in (0,\infty]$, the $r$-th modulus of smoothness of $f$ is defined by*

$$w_{r,p}(f,t) = \sup_{h \in \mathbb{R}^d : \|h\|_2 \leq t} \|\Delta_h^r(f)\|_p,$$

*where $\Delta_h^r(f)(x) = \begin{cases} \sum_{j=0}^r \binom{r}{j}(-1)^{r-j} f(x+jh) & (x \in \Omega, \; x+rh \in \Omega), \\ 0 & (otherwise). \end{cases}$*

Based on the modulus of smoothness, the Besov space is defined as in the following definition.

**Definition 4** (Besov space ($B_{p,q}^\alpha(\Omega)$)). *For $0 < p, q \leq \infty$, $\alpha > 0$, $r := \lfloor \alpha \rfloor + 1$, let the seminorm $|\cdot|_{B_{p,q}^\alpha}$ be*

$$|f|_{B_{p,q}^\alpha} := \begin{cases} \left(\int_0^\infty (t^{-\alpha} w_{r,p}(f,t))^q \frac{dt}{t}\right)^{\frac{1}{q}} & (q < \infty), \\ \sup_{t>0} t^{-\alpha} w_{r,p}(f,t) & (q = \infty). \end{cases}$$

*The norm of the Besov space $B_{p,q}^\alpha(\Omega)$ can be defined by $\|f\|_{B_{p,q}^\alpha} := \|f\|_p + |f|_{B_{p,q}^\alpha}$, and $B_{p,q}^\alpha(\Omega) = \{f \in L^p(\Omega) \mid \|f\|_{B_{p,q}^\alpha} < \infty\}$.*

---

[2] Since $\Omega = [0,1]^d$ in our setting, the boundedness automatically follows from the continuity.

[3] We let $\mathbb{N} := \{1,2,3,\ldots\}$, $\mathbb{Z}_+ := \{0,1,2,3,\ldots\}$, $\mathbb{Z}_+^d := \{(z_1,\ldots,z_d) \mid z_i \in \mathbb{Z}_+\}$, $\mathbb{R}_+ := \{x \geq 0 \mid x \in \mathbb{R}\}$, and $\mathbb{R}_{++} := \{x > 0 \mid x \in \mathbb{R}\}$.

Note that $p, q < 1$ is also allowed. In that setting, the Besov space is no longer a Banach space but a quasi-Banach space. The Besov space plays an important role in several fields such as nonparametric statistical inference (Kerkyacharian & Picard, 1992; Donoho et al., 1996; Donoho & Johnstone, 1998; Giné & Nickl, 2015) and approximation theory (Temlyakov, 1993a). These spaces are closely related to each other as follows (Triebel, 1983):

- For $m \in \mathbb{N}$, $B_{p,1}^m(\Omega) \hookrightarrow W_p^m(\Omega) \hookrightarrow B_{p,\infty}^m(\Omega)$, and $B_{2,2}^m(\Omega) = W_2^m(\Omega)$ [4].

- For $0 < s < \infty$ and $s \notin \mathbb{N}$, $\mathcal{C}^s(\Omega) = B_{\infty,\infty}^s(\Omega)$.

- For $0 < s, p, q, r \leq \infty$ with $s > \delta := d(1/p - 1/r)_+$, it holds that $B_{p,q}^s(\Omega) \hookrightarrow B_{r,q}^{s-\delta}(\Omega)$. In particular, under the same condition, from the definition of $\|\cdot\|_{B_{p,q}^s}$, it holds that

$$B_{p,q}^s(\Omega) \hookrightarrow L^r(\Omega). \tag{1}$$

- For $0 < s, p, q \leq \infty$, if $s > d/p$, then

$$B_{p,q}^s(\Omega) \hookrightarrow \mathcal{C}^0(\Omega). \tag{2}$$

Hence, if the smoothness parameter satisfies $s > d/p$, then it is continuously embedded in the set of the continuous functions. However, if $s < d/p$, then the elements in the space are no longer continuous. Moreover, it is known that $B_{1,1}^1([0,1])$ is included in the space of bounded total variation and $B_{1,\infty}^1([0,1])$ includes it (Peetre & Dept, 1976). Hence, the Besov space also allows spatially inhomogeneous smoothness with spikes and jumps; which makes difference between linear estimators and deep learning (see Sec. 4.1).

It is known that the minimax rate to estimate $f^\circ$ is lower bounded by $n^{-2s/(2s+d)}$, (Kerkyacharian & Picard, 1992; Donoho et al., 1996; Donoho & Johnstone, 1998; Giné & Nickl, 2015). We see that the *curse of dimensionality* is unavoidable as long as we consider the Besov space. This is an undesirable property because we easily encounter high dimensional data in several machine learning problems. Hence, we need another condition to derive approximation and estimation error bounds that are not heavily affected by the dimensionality. To do so, we introduce the notion of *mixed smoothness*. The Besov space with mixed smoothness is defined as follows (Schmeisser, 1987; Sickel & Ullrich, 2009). To define the space, we define the coordinate difference operator as

$$\Delta_h^{r,i}(f)(x) = \Delta_h^r(f(x_1, \ldots, x_{i-1}, \cdot, x_{i+1}, \ldots, x_d))(x_i)$$

for $f : \mathbb{R}^d \to \mathbb{R}$, $h \in \mathbb{R}_+$, $i \in [d]$, and $r \geq 1$. By applying this difference operator to each coordinate in a subset $e \subset \{1, \ldots, d\}$ of coordinates, the mixed differential operator for a step length $h \in \mathbb{R}^d$ is defined as

$$\Delta_h^{r,e}(f) = \left( \prod_{i \in e} \Delta_{h_i}^{r,i} \right)(f), \quad \Delta_h^{r,\emptyset}(f) = f.$$

Then, the mixed modulus of smoothness is defined as

$$w_{r,p}^e(f, t) := \sup_{|h_i| \leq t_i, i \in e} \|\Delta_h^{r,e}(f)\|_p$$

for $t \in \mathbb{R}_+^d$ and $0 < p \leq \infty$. This quantity measures how the function $f$ is "rough" in a coordinate-wise manner. In contrast to the Besov space (Def. 3), the differentiation is taken over all coordinates in $e$, which imposes a coordinate-wise smoothness. Letting $0 < p, q \leq \infty$, $\alpha \in \mathbb{R}_{++}^d$ and $r_i := \lfloor \alpha_i \rfloor + 1$, the semi-norm $|\cdot|_{MB_{p,q}^{\alpha,e}}$ based on the mixed smoothness is defined by

$$|f|_{MB_{p,q}^{\alpha,e}} := \begin{cases} \left\{ \int_\Omega [(\prod_{i \in e} t_i^{-\alpha_i}) w_{r,p}^e(f,t)]^q \frac{\mathrm{d}t}{\prod_{i \in e} t_i} \right\}^{1/q} & (0 < q < \infty), \\ \sup_{t \in \Omega} (\prod_{i \in e} t_i^{-\alpha_i}) w_{r,p}^e(f,t) & (q = \infty). \end{cases}$$

By summing up the semi-norm over the choice of $e$, the (quasi-)norm of the mixed smooth Besov space (abbreviated to m-Besov space) is defined by

$$\|f\|_{MB_{p,q}^\alpha} := \|f\|_p + \sum_{e \subset \{1, \ldots, d\}} |f|_{MB_{p,q}^{\alpha,e}},$$

---

[4] The notation $\hookrightarrow$ means a continuous embedding; that is, $X \hookrightarrow Y$ for two normed spaces $X, Y$ means $X$ is continuously embedded in $Y$.

and thus $MB^\alpha_{p,q}(\Omega) := \{f \in L^p(\Omega) \mid \|f\|_{MB^\alpha_{p,q}} < \infty\}$ where $0 < p, q \leq \infty$ and $\alpha \in \mathbb{R}^d_{++}$. In this paper, we assume that $\alpha_1 = \cdots = \alpha_d$. With a slight abuse of notation, we also use the notation $MB^\alpha_{p,q}$ for $\alpha > 0$ to indicate $MB^{(\alpha,\dots,\alpha)}_{p,q}$.

It is known that, when $p = q$, the m-Besov space is characterized as a *tensor product space* of $B^s_{p,q}([0,1])$ (Sickel & Ullrich, 2009). The m-Besov space includes several important models considered in the literature of statistical learning, e.g., the additive model (Meier et al., 2009) and the tensor model (Signoretto et al., 2010). It is known that an appropriate estimator in these models can avoid curse of dimensionality (Meier et al., 2009; Raskutti et al., 2012; Kanagawa et al., 2016; Suzuki et al., 2016). What we will show in this paper supports that this fact is also applied to deep learning from a unifying viewpoint.

The difference between the (normal) Besov space and the m-Besov space can be informally explained as follows. For regularity condition $\alpha_i \leq 2$ ($i = 1, 2$), the m-Besov space consists of functions for which the following derivatives are "bounded":

$$\frac{\partial f}{\partial x_1}, \frac{\partial f}{\partial x_2}, \frac{\partial^2 f}{\partial x_1^2}, \frac{\partial^2 f}{\partial x_2^2}, \frac{\partial^2 f}{\partial x_1 \partial x_2}, \frac{\partial^3 f}{\partial x_1 \partial x_2^2}, \frac{\partial^3 f}{\partial x_1^2 \partial x_2}, \frac{\partial^4 f}{\partial x_1^2 \partial x_2^2}.$$

That is, the "max" of the orders of derivatives over coordinates needs to be bounded by 2. On the other hand, the Besov space only ensures the boundedness of the following derivatives:

$$\frac{\partial f}{\partial x_1}, \frac{\partial f}{\partial x_2}, \frac{\partial^2 f}{\partial x_1^2}, \frac{\partial^2 f}{\partial x_2^2}, \frac{\partial^2 f}{\partial x_1 \partial x_2},$$

where the "sum" of the orders needs to be bounded by 2. This difference directly affects the rate of convergence of approximation accuracy. Further details about this space and related topics can be found in a comprehensive survey (Dũng et al., 2016).

**Relation to Barron class.** Barron (1991; 1993; 1994) showed that, if the Fourier transform of a function $f$ satisfies some integrability condition, then we may avoid curse of dimensionality for estimating neural networks with sigmoidal activation functions. The integrability condition is given by $\int_{\mathbb{C}^d} \|\omega\| |\hat{f}(\omega)| \mathrm{d}\omega < \infty$, where $\hat{f}$ is the Fourier transform of a function $f$. We call the class of functions satisfying this condition *Barron class*. A similar function class is analyzed by Klusowski & Barron (2016) too. We cannot compare directly the m-Besov space with the Barron class, but they are closely related. Indeed, if $p = q = 2$ and $s = \alpha_1 = \cdots = \alpha_d$, then m-Besov space $MB^s_{2,2}(\Omega)$ is equivalent to the tensor product of Sobolev space (Sickel & Ullrich, 2011) which consists of functions $f : \Omega \to \mathbb{R}$ satisfying $\int_{\mathbb{C}^d} \prod_{i=1}^d (1 + |\omega_i|^2)^s |\hat{f}(\omega)|^2 \mathrm{d}\omega < \infty$. Therefore, our analysis gives a (similar but) different characterization of conditions to avoid curse of dimensionality.

## 3 APPROXIMATION ERROR ANALYSIS

In this section, we evaluate how well the functions in the Besov and m-Besov spaces can be approximated by neural networks with the ReLU activation. Let us denote the ReLU activation by $\eta(x) = \max\{x, 0\}$ ($x \in \mathbb{R}$), and for a vector $x$, $\eta(x)$ is operated in an element-wise manner. Define the neural network with height $L$, width $W$, sparsity constraint $S$ and norm constraint $B$ as

$$\Phi(L, W, S, B) := \{(\mathcal{W}^{(L)} \eta(\cdot) + b^{(L)}) \circ \cdots \circ (\mathcal{W}^{(1)} x + b^{(1)}) \mid \mathcal{W}^{(L)} \in \mathbb{R}^{1 \times W}, \ b^{(L)} \in \mathbb{R},$$

$$\mathcal{W}^{(1)} \in \mathbb{R}^{W \times d}, \ b^{(1)} \in \mathbb{R}^W, \ \mathcal{W}^{(\ell)} \in \mathbb{R}^{W \times W}, \ b^{(\ell)} \in \mathbb{R}^W (1 < \ell < L),$$

$$\sum_{\ell=1}^L (\|\mathcal{W}^{(\ell)}\|_0 + \|b^{(\ell)}\|_0) \leq S, \max_\ell \|\mathcal{W}^{(\ell)}\|_\infty \vee \|b^{(\ell)}\|_\infty \leq B\},$$

where $\|\cdot\|_0$ is the $\ell_0$-norm of the matrix (the number of non-zero elements of the matrix) and $\|\cdot\|_\infty$ is the $\ell_\infty$-norm of the matrix (maximum of the absolute values of the elements). We want to evaluate how large $L, W, S, B$ should be to approximate $f^o \in MB^\alpha_{p,q}(\Omega)$ by an element $f \in \Phi(L, W, S, B)$ with precision $\epsilon > 0$ measured by $L^r$-norm: $\min_{f \in \Phi} \|f - f^o\|_r \leq \epsilon$.

### 3.1 APPROXIMATION ERROR ANALYSIS FOR BESOV SPACES

Here, we show how the neural network can approximate a function in the Besov space which is useful to derive the generalization error of deep learning. Although its derivation is rather standard

as considered in Chui et al. (1994); Bölcskei et al. (2017), it should be worth noting that the bound derived here cannot be attained by any *non-adaptive* method and the generalization error based on the analysis is also unattainable by any *linear* estimators including the kernel ridge regression. That explains the high adaptivity of deep neural network and how it outperforms usual linear methods such as kernel methods.

To show the approximation accuracy, a key step is to show that the ReLU neural network can approximate the *cardinal B-spline* with high accuracy. Let $\mathcal{N}(x) = 1 \ (x \in [0,1])$, $0$ (otherwise), then the *cardinal B-spline of order* $m$ is defined by taking $m + 1$-times convolution of $\mathcal{N}$:

$$\mathcal{N}_m(x) = (\underbrace{\mathcal{N} * \mathcal{N} * \cdots * \mathcal{N}}_{m + 1 \text{ times}})(x),$$

where $f * g(x) := \int f(x - t)g(t)\mathrm{d}t$. It is known that $\mathcal{N}_m$ is a piece-wise polynomial of order $m$. For $k = (k_1, \ldots, k_d) \in \mathbb{Z}_+^d$ and $j = (j_1, \ldots, j_d) \in \mathbb{Z}^d$, let $M_{k,j}^d(x) = \prod_{i=1}^d \mathcal{N}_m(2^{k_i} x_i - j_i)$. Even for $k \in \mathbb{Z}_+$, we also use the same notation to express $M_{k,j}^d(x) = \prod_{i=1}^d \mathcal{N}_m(2^k x_i - j_i)$. Here, $k$ controls the spatial "resolution" and $j$ specifies the location on which the basis is put. Basically, we approximate a function $f$ in a Besov space by a super-position of $M_{k,j}^m(x)$, which is closely related to wavelet analysis (Mallat, 1999).

Mhaskar & Micchelli (1992); Chui et al. (1994) have shown the approximation ability of neural network for a function with bounded modulus of smoothness. However, the class of the activation functions in their analysis does not include ReLU but they dealt with activation functions satisfying the following conditions,

$$\lim_{x \to \infty} \eta(x)/x^k \to 1, \quad \lim_{x \to -\infty} \eta(x)/x^k = 0, \ \exists K > 1 \text{ s.t. } |\eta(x)| \le K(1 + |x|)^k \ (x \in \mathbb{R}), \quad (3a)$$

for $k = 2$ which excludes ReLU. Mhaskar (1993) analyzed deep neural network under the same setting but it restricts the smoothness parameter to $s = k + 1$. Mhaskar (1996) considered the Sobolev space $W_p^m$ with an infinitely many differentiable "bump" function which also excludes ReLU. However, approximating the cardinal B-spline by ReLU can be attained by appropriately using the technique developed by Yarotsky (2017) as in the following lemma.

**Lemma 1** (Approximation of cardinal B-spline basis by the ReLU activation). *There exists a constant $c_{(d,m)}$ depending only on $d$ and $m$ such that, for all $\epsilon > 0$, there exists a neural network $\check{M} \in \Phi(L_0, W_0, S_0, B_0)$ with $L_0 := 3 + 2\left\lceil \log_2\left(\frac{3^{d \vee m}}{\epsilon c_{(d,m)}}\right) + 5\right\rceil \lceil \log_2(d \vee m)\rceil$, $W_0 := 6dm(m+2)+2d$, $S_0 := L_0 W_0^2$ and $B_0 := 2(m+1)^m$ that satisfies*

$$\|M_{0,0}^d - \check{M}\|_{L^\infty(\mathbb{R}^d)} \le \epsilon,$$

*and $\check{M}(x) = 0$ for all $x \notin [0, m+1]^d$.*

The proof is in Appendix A. Based on this lemma, we can translate several B-spline approximation results into those of deep neural network approximation. In particular, combining this lemma and the B-spline interpolant representations of functions in Besov spaces (DeVore & Popov, 1988; DeVore et al., 1993; Dũng, 2011b), we obtain the optimal approximation error bound for deep neural networks. Here, let $U(\mathcal{H})$ be the unit ball of a quasi-Banach space $\mathcal{H}$, and for a set $\mathcal{F}$ of functions, define the worst case approximation error as

$$R_r(\mathcal{F}, \mathcal{H}) := \sup_{f^\circ \in U(\mathcal{H})} \inf_{f \in \mathcal{F}} \|f^\circ - f\|_{L^r([0,1]^d)}.$$

**Proposition 1** (Approximation ability for Besov space). *Suppose that $0 < p, q, r \le \infty$ and $0 < s < \infty$ satisfy the following condition:*

$$s > d(1/p - 1/r)_+. \quad (4)$$

*Assume that $m \in \mathbb{N}$ satisfies $0 < s < \min(m, m - 1 + 1/p)$. Let $\delta = d(1/p - 1/r)_+$ and $\nu = (s - \delta)/(2\delta)$. For sufficiently large $N \in \mathbb{N}$ and $\epsilon = N^{-s/d - (\nu^{-1} + d^{-1})(d/p - s)_+} \log(N)^{-1}$, let*

$$L = 3 + 2\lceil \log_2\left(\frac{3^{d \vee m}}{\epsilon c_{(d,m)}}\right) + 5\rceil \lceil \log_2(d \vee m)\rceil, \qquad W = NW_0,$$

$$S = [(L-1)W_0^2 + 1]N, \qquad\qquad B = O(N^{(\nu^{-1} + d^{-1})(1 \vee (d/p - s)_+)}),$$

*then it holds that*

$$R_r(\Phi(L, W, S, B), B_{p,q}^s([0,1]^d)) \lesssim N^{-s/d}.$$

**Remark 1.** *By Eq. (1), the condition (4) indicates that $f^{\circ} \in B_{p,q}^s$ satisfies $f^{\circ} \in L^r(\Omega)$. If we set $p = q = \infty$ and $r = \infty$, then $B_{p,q}^s(\Omega) = C^s(\Omega)$ which yields the result by Yarotsky (2017) as a special case.*

The proof is in Appendix B. According to the theorem, the approximation error $N^{-s/d}$ can be achieved by the settings $L = O(\log(N))$, $W = O(N)$ and $S = O(N \log(N))$ for $N \in \mathbb{N}$. The convergence rate $N^{-s/d}$ is controlled by the smoothness $s$ and the dimensionality $d$; as the smoothness $s$ goes up, we have better approximation error, and as the dimensionality $d$ goes up, we have worse error. An interesting point is that the statement is valid even for $p \neq r$. In particular, the theorem also supports non-continuous regime ($s < d/p$ in which $L^{\infty}$-convergence does no longer hold but instead $L^r$-convergence is guaranteed under the condition $s > d(1/p - 1/r)_+$. In that sense, the convergence of the approximation error is guaranteed in considerably general settings. Pinkus (1999) gave an explicit form of convergence when $1 \leq p = r$ for the activation functions satisfying Eq. (3) which does not cover ReLU and an important setting $p \neq r$. Petrushev (1998) considered $p = r = 2$ and activation function with Eq. (3) where $s$ is an integer such that $s \leq k+1+(d-1)/2$. Chui et al. (1994) and Bölcskei et al. (2017) dealt with the smooth sigmoidal activation satisfying the condition (3) with $k \geq 2$ or a "smoothed version" of the ReLU activation which excludes ReLU; but Bölcskei et al. (2017) presented a general strategy for neural-net approximation by using the notion of best $M$-term approximation. Mhaskar & Micchelli (1992) gives an approximation bound using the modulus of smoothness, but the smoothness $s$ and the order of sigmoidal function $k$ in (3) is tightly connected and $f^{\circ}$ is assumed to be continuous which excludes the situation $s < d/p$. On the other hand, the above proposition does not require such a tight connection and it explicitly gives the approximation bound for Besov spaces. Williamson & Bartlett (1992) derived a spline approximation error bound for an element in a Besov space when $d = 1$, but the derived bound is only $O(N^{-s+(1/p-1/r)_+})$ which is the rate of non-adaptive methods described below, and approximation by a ReLU activation network is not discussed. We may also use the analysis of Cohen et al. (2001) which is based on compactly supported wavelet bases, but the cardinal B-spline is easy to handle through quasi-interpolant representation as performed in the proof of Proposition 1.

It should be noted that the presented approximation accuracy bound is not trivial because it can not be achieved by a *non-adaptive method*. Actually, the *best $N$-term approximation error* (Kolmorogov width) of the Besov space is lower bounded as

$$
\inf_{S_N \subset B_{p,q}^s} \sup_{f \in U(B_{p,q}^s)} \inf_{\check{f} \in S_N} \|f - \check{f}\|_{L^r(\Omega)} \gtrsim \begin{cases} N^{-s/d+(1/p-1/r)_+} & (1 < p < r \leq 2, \ s > d(1/p - 1/r)), \\ N^{-s/d+1/p-1/2} & (1 < p < 2 < r \leq \infty, \ s > d/p), \\ N^{-s/d} & (2 \leq p < r \leq \infty, \ s > d/2), \end{cases}
$$

(5)

if $1 < p < r \leq \infty$, $1 \leq q < \infty$ and $1 < s$, where $S_N$ is any $N$-dimensional subspace of $B_{p,q}^s$ (Romanyuk, 2009; Myronyuk, 2016; Vybáral, 2008). That is, any linear/non-linear approximator with *fixed $N$-bases* does not achieve the approximation error $N^{-s/d}$ in some parameter settings such as $1 < p < 2 < r$. On the other hand, adaptive methods including deep learning can improve the error rate up to $N^{-s/d}$ which is rate optimal (Dũng, 2011b). The difference is significant when $p < r$. This implies that deep neural network possesses high adaptivity to find which part of the function should be intensively approximated. In other words, deep neural network can properly extracts the feature of the input (which corresponds to construct an appropriate set of bases) to approximate the target function in the most efficient way.

### 3.2 APPROXIMATION ERROR ANALYSIS FOR M-BESOV SPACE

Here, we deal with m-Besov spaces instead of the ordinary Besov space. The next theorem gives the approximation error bound to approximate functions in the m-Besov spaces by deep neural network models. Define $D_{k,d} := \left(1 + \frac{d-1}{k}\right)^k \left(1 + \frac{k}{d-1}\right)^{d-1}$. Then, we have the following theorem.

**Theorem 1** (Approximation ability for m-Besov space)**.** *Suppose that $0 < p, q, r \leq \infty$ and $s < \infty$ satisfies $s > (1/p - 1/r)_+$. Assume that $m \in \mathbb{N}$ satisfies $0 < s < \min(m, m - 1 + 1/p)$. Let $\delta = (1/p - 1/r)_+$ and $\nu = (s - \delta)/(2\delta)$. For any $K \geq 1$, let $K^* = \lceil K(1 + \frac{2\delta}{\alpha - \delta}) \rceil$. Then, for $N = (2 + (1 - 2^{-\nu})^{-1})2^K D_{K^*,d}$, if we set*

$$L = 3 + 2 \left\lceil \log_2\left(\frac{3^{d \vee m}}{c_{(d,m)}}\right) + 5 + (s + (\frac{1}{p} - s)_+ + 1)K^* + \log([e(m+1)]^d(1 + K^*))\right\rceil \lceil \log_2(d \vee m) \rceil,$$

$$W = W_0 N, \ \ S = [(L-1)W_0^2 + 1]N, \ \ B = O(N^{(\nu^{-1}+1)(1\vee(1/p-s)_+)}),$$

*then it holds that*

*(i) For $p \geq r$,*

$$R_r(\Phi(L,W,S,B), MB^s_{p,q}([0,1]^d)) \lesssim 2^{-Ks} D_{K,d}^{(1/\min(r,1)-1/q)_+}, \tag{6a}$$

*(ii) For $p < r$,*

$$R_r(\Phi(L,W,S,B), MB^s_{p,q}([0,1]^d)) \lesssim \begin{cases} 2^{-Ks} D_{K,d}^{(1/r-1/q)_+} & (r < \infty), \\ 2^{-Ks} D_{K,d}^{(1-1/q)_+} & (r = \infty). \end{cases} \tag{6b}$$

The proof is given in Appendix C.3. It holds that $N \simeq 2^K K^{(d-1)}$, which implies $2^{-K} \simeq N^{-1} \log^{d-1}(N)$ if $N \gg d$ (see also the discussion right after Theorem 5 in Appendix C.1 for more details of calculation). Therefore, when $r \gg q$, the approximation error can be evaluated as $O(N^{-s} \log^{s(d-1)}(N))$ for $L = O(\log(N))$, $W = O(N)$ and $S = O(N \log(N))$ for $N \in \mathbb{N}$ in which the effect of dimensionality $d$ is much milder than that of Proposition 1. This means that the curse of dimensionality is much eased in the mixed smooth space.

The obtained bound is far from obvious. Actually, it is better than any linear approximation methods as follows. Let the linear $M$-width introduced by Tikhomirov (1960) be $\lambda_N(MB^s_{p,q}, L^r) := \inf_{L_N} \sup_{f \in U(MB^s_{p,q})} \|f - L_N(f)\|_r$, where the infimum is taken over all linear oprators $L_N$ with rank $N$ from $MB^s_{p,q}$ to $L^r$. The linear $N$-width of the m-Besov space has been extensively studied as in the following proposition (see Lemma 5.1 of Dũng (2011a), and Romanyuk (2001)).

**Proposition 2.** *Let $1 \leq p, r \leq \infty$, $0 < q \leq \infty$ and $s > (1/p - 1/r)_+$. Then we have the following asymptotic order of the linear width for the asymptotics $N \gg d$:*
*(a) For $p \geq r$,*

$$\lambda_N(MB^s_{p,q}, L^r) \simeq \begin{cases} (N^{-1} \log^{d-1}(N))^s & \begin{cases} (q \leq 2 \leq r \leq p < \infty), \\ (q \leq 1, \ p = r = \infty), \\ (1 < p = r \leq 2, \ q \leq r), \end{cases} \\ (N^{-1} \log^{d-1}(N))^s (\log^{d-1}(N))^{1/r-1/q} & (1 < p = r \leq 2, \ q > r), \\ (N^{-1} \log^{d-1}(N))^s (\log^{d-1}(N))^{(1/2-1/q)_+} & (2 \leq q, \ 1 < r < 2 \leq p < \infty), \end{cases}$$

*(b) For $1 < p < r < \infty$,*

$$\lambda_N(MB^s_{p,q}, L^r) \simeq \begin{cases} (N^{-1} \log^{d-1}(N))^{s+1/r-1/p} & (2 \leq p, \ 2 \leq q \leq r), \\ (N^{-1} \log^{d-1}(N))^{s+1/r-1/p} (\log^{d-1}(N))^{(1/r-1/q)_+} & (r \leq 2). \end{cases}$$

Therefore, the approximation error given in Theorem 1 achieves the optimal linear width $((N^{-1} \log^{d-1}(N))^s)$ for several parameter settings of $p, q, s$. In particular, when $p < r$, the bound in Theorem 1 is better than that of Proposition 2. This is because to prove Theorem 1, we used an adaptive recovery technique instead of a linear recovery method. This implies that, by constructing a deep neural network accurately, we achieve the same approximation accuracy as the adaptive one which is better than that of linear approximation.

## 4 ESTIMATION ERROR ANALYSIS

In this section, we connect the approximation theory to generalization error analysis (estimation error analysis). For the statistical analysis, we assume the following nonparametric regression model:

$$y_i = f^o(x_i) + \xi_i \quad (i = 1, \dots, n),$$

where $x_i \sim P_X$ with density $0 \leq p(x) < R$ on $[0,1]^d$, and $\xi_i \sim N(0, \sigma^2)$. The data $D_n = (x_i, y_i)_{i=1}^n$ is independently identically distributed. We want to estimate $f^o$ from the data. Here, we consider a regularized learning procedure:

$$\hat{f} = \operatorname*{argmin}_{\bar{f}: f \in \Phi(L,W,S,B)} \sum_{i=1}^n (y_i - \bar{f}(x_i))^2 \tag{7}$$

where $\bar{f}$ is the *clipping* of $f$ defined by $\bar{f} = \min\{\max\{f, -F\}, F\}$ for $F > 0$ which is realized by ReLU units. Since the sparsity level is controlled by $S$ and the parameter is bounded by $B$, this estimator can be regarded as a regularized estimator. In practice, it is hard to exactly compute $\hat{f}$. Thus, we approximately solve the problem by applying sparse regularization such as $L_1$-regularization and optimal parameter search through Bayesian optimization. The generalization error that we present here is an "ideal" bound which is valid if the optimal solution $\hat{f}$ is computable.

## 4.1 ESTIMATION ERROR IN BESOV SPACES

In this subsection, we provide the estimation error rate of deep learning to estimate functions in Besov spaces.

**Theorem 2.** *Suppose that $0 < p, q \leq \infty$ and $s > d(1/p - 1/2)_+$. If $f^\circ \in B_{p,q}^s(\Omega) \cap L^\infty(\Omega)$ and. $\|f^\circ\|_{B_{p,q}^s} \leq 1$ and $\|f^\circ\|_\infty \leq F$ for $F \geq 1$, then letting $(W, L, S, B)$ be as in Proposition 1 with $N \asymp n^{\frac{d}{2s+d}}$, we obtain*

$$\mathrm{E}_{D_n}[\|f^\circ - \widehat{f}\|_{L^2(P_X)}^2] \lesssim n^{-\frac{2s}{2s+d}} \log(n)^3,$$

*where $\mathrm{E}_{D_n}[\cdot]$ indicates the expectation w.r.t. the training data $D_n$.*

The proof is given in Appendix D. The condition $\|f^\circ\|_\infty \leq F$ is required to connect the empirical $L^2$-norm $\frac{1}{n} \sum_{i=1}^n (\widehat{f}(x_i) - f^\circ(x_i))^2$ to the population $L^2$-norm $\|\widehat{f} - f^\circ\|_{L^2(P_X)}^2$. It is known that the convergence rate $n^{-\frac{2s}{2s+d}}$ is mini-max optimal (Kerkyacharian & Picard, 1992; Donoho et al., 1996; Donoho & Johnstone, 1998; Giné & Nickl, 2015). Thus, it cannot be improved by any estimator. Therefore, deep learning can achieve the minimax optimal rate up to $\log(n)^3$-order. The term $\log(n)^3$ could be improved to $\log(n)^2$ by using the construction of Petersen & Voigtlaender (2017). However, we don't pursue this direction for simplicity.

Here an important remark is that this minimax optimal rate cannot be achieved by any *linear estimator*. We call an estimator *linear* when the estimator depends on $(y_i)_{i=1}^n$ linearly (it can be non-linearly dependent on $(x_i)_{i=1}^n$). Several classical methods such as the kernel ridge regression, the Nadaraya-Watson estimator and the sieve estimator are included in the class of linear estimators (e.g., kernel ridge regression is given as $\widehat{f}(x) = k_{x,X}(k_{XX} + \lambda \mathrm{I})^{-1}Y$). The following proposition given by Donoho & Johnstone (1998); Zhang et al. (2002) states that the minimax rate of linear estimators is lower bounded by $n^{-\{2s-2(1/p-1/2)_+\}/\{2s+1-2(1/p-1/2)_+\}}$ for $d = 1$ which is larger than the minimax rate $n^{-\frac{2s}{2s+1}}$ if $p < 2$.

**Proposition 3** (Donoho & Johnstone (1998); Zhang et al. (2002)). *Suppose that $d = 1$ and the input distribution $P_X$ is the uniform distribution on $[0, 1]$. Assume that $s > 1/p$, $1 \leq p, q \leq \infty$ or $s = p = q = 1$. Then,*

$$\inf_{\widehat{f}: \text{ linear}} \sup_{f^\circ \in U(B_{p,q}^s)} \mathrm{E}_{D_n}[\|f^\circ - \widehat{f}\|_{L^2(P_X)}^2] \gtrsim n^{-\frac{2s-v}{2s+1-v}}$$

*where $v = 2/(p \wedge 2) - 1$ and $\widehat{f}$ runs over all linear estimators, that is, $\widehat{f}$ depends on $(y_i)_{i=1}^n$ linearly.*

When $p < 2$, the smoothness of the Besov space is somewhat inhomogeneous, that is, a function in the Besov space contains spiky/jump parts and smooth parts (remember that when $s = p = q = 1$ for $d = 1$, the Besov space is included in the set of functions with bounded total variation). Here, the setting $p < 2$ is the regime where there appears difference between non-adaptive methods and deep learning in terms of approximation accuracy (see Eq. (5)). On the other hand, the linear estimator captures only global properties of the function and cannot capture variability of local shapes of the function. Hence, the linear estimator cannot achieve the minimax optimal rate if the function has spatially inhomogeneous smoothness. However, deep learning possesses adaptivity to the spatial inhomogeneity.

Imaizumi & Fukumizu (2018) has pointed out that such a discrepancy appears when the target function is *non-smooth*. Interestingly, the parameter setting $s > 1/p$ assumed in Proposition 3 ensures smoothness (see Eq. (2)). This means that non-smoothness is not necessarily required to characterize the superiority of deep learning, but *non-convexity* of the set of target functions is essentially important. In fact, the gap is coming from the property that the *quadratic hull* of the model $U(B_{p,q}^s)$ is strictly larger than the original set (Donoho & Johnstone, 1998).

## 4.2 ESTIMATION ERROR IN MIXED SMOOTH BESOV SPACES

Here, we provide the estimation error rate to estimate functions in mixed smooth Besov spaces.

**Theorem 3.** *Suppose that $0 < p, q \leq \infty$ and $s > (1/p - 1/2)_+$. Let $u = (1 - 1/q)_+$ for $p \geq 2$ and $u = (1/2 - 1/q)_+$ for $p < 2$. If $f^\circ \in MB_{p,q}^s(\Omega) \cap L^\infty(\Omega)$ and $\|f^\circ\|_{MB_{p,q}^s} \leq 1$ and $\|f^\circ\|_\infty \leq F$*

*for $F \geq 1$, then letting $(W, L, S, B)$ be as in Theorem 1, we obtain*

$$\mathrm{E}_{D_n}[\|f^{\mathrm{o}} - \widehat{f}\|_{L^2(P_X)}^2] \lesssim n^{-\frac{2s}{2s+1}} \log(n)^{\frac{2(d-1)(u+s)}{1+2s}} \log(n)^3.$$

*Under the same assumption, if $s > u \log_2(e)$ is additionally satisfied, we also have*

$$\mathrm{E}_{D_n}[\|f^{\mathrm{o}} - \widehat{f}\|_{L^2(P_X)}^2] \lesssim n^{-\frac{2s-2u\log_2(e)}{2s+1+(1-2u)\log_2(e)}} \log(n)^3.$$

The proof is given in Appendix D. The risk bound (Theorem 3) indicates that the curse of dimensionality can be eased by assuming the mixed smoothness compared with the ordinary Besov space ($n^{-\frac{2s}{2s+d}}$). We show that this is almost minimax optimal in Theorem 4 below. In the first bound, the dimensionality $d$ comes in the exponent of $\mathrm{poly} \log(n)$ term. If $u = 0$, then the effect of $d$ can be further eased. Actually, in this situation ($u = 0$), the second bound can be rewritten as

$$n^{-\frac{2s}{2s+1+\log_2(e)}} \log(n)^3,$$

where the effect of the dimensionality $d$ completely disappears from the exponent. This explains partially why deep learning performs well for high dimensional data. Here, we again remark the adaptivity of deep learning. Remind that this rate cannot be achieved by the linear estimator for $p < 2$ when $d = 1$ by Proposition 3. Montanelli & Du (2017) has analyzed the mixed smooth Hölder space with $s < 2$. However, our analysis is applicable to the m-Besov space which is more general than the mixed smooth Hölder space and the covered range of $s, p, q$ is much larger.

**Minimax optimal rate for estimating a function in the m-Besov space**   Here, we show the minimax optimality of the obtained bound as follows.

**Theorem 4.** *Assume that $0 < p, q \leq \infty$ and $s > (1/p - 1/2)_+$ and $P_X$ is the uniform distribution over $[0,1]^d$. Regarding $d$ as a constant, the minimax learning rate in the asymptotics of $n \to \infty$ is lower bounded as follows: There exists a constant $\widehat{C}_1$ such that*

$$\inf_{\widehat{f}} \sup_{f^{\mathrm{o}} \in U(MB_{p,q}^s)} \mathrm{E}_{D_n}[\|\widehat{f} - f^{\mathrm{o}}\|_{L^2(P_X)}^2] \geq \widehat{C}_1 n^{-\frac{2s}{2s+1}} \log(n)^{\frac{2(d-1)(s+1/2-1/q)_+}{2s+1}} \tag{8}$$

*where "inf" is taken over all measurable functions of the observations $(x_i, y_i)_{i=1}^n$ and the expectation is taken for the sample distribution.*

The proof is given in Appendix E. Because of this theorem, our bound given in Theorem 3 achieves the minimax optimal rate in the regime of $p < 2$ and $1/2 - 1/q > 0$ up to $\log(n)^3$ order. Even outside of this parameter setting, the discrepancy between our upper bound and the minimax lower bound is just a poly-$\log$ oder. See also Neumann (2000) for some other related spaces and specific examples such as $p = q = 2$.

## 5   CONCLUSION

This paper investigated the learning ability of deep ReLU neural network when the target function is in a Besov space or a mixed smooth Besov space. Based on the analysis for the Besov space, it was shown that deep learning using the ReLU activation can achieve the minimax optimal rate and outperform the linear method when $p < 2$ which indicates the spatial inhomogeneity of the shape of the target function. The analysis for the mixed smooth Besov space showed that deep learning can adaptively avoid the curse of dimensionality. The bound was derived by sparse grid technique. All analyses in the paper adopted the cardinal B-spline expansion and the adaptive non-linear approximation technique, which allowed us to show the minimax optimal rate. The consequences of the analyses partly support the superiority of deep leaning in terms of adaptivity and ability to avoid curse of dimensionality. From more high level view point, these favorable property is reduced to its high feature extraction ability.

**Acknowledgment**   The author is grateful to Satoshi Hayakawa for showing the correct statement of Proposition 4 by modifying the proof of Lemma 10 of Schmidt-Hieber (2018). TS was partially supported by MEXT Kakenhi (25730013, 25120012, 26280009, 15H05707 and 18H03201), Japan Digital Design and JST-CREST.

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

## A  PROOF OF LEMMA 1

*Proof of Lemma 1.* First note that $\mathcal{N}_m(x) = \frac{1}{m!}\sum_{j=0}^{m+1}(-1)^j\binom{m+1}{j}(x-j)_+^m$ (see Eq. (4.28) of Mhaskar & Micchelli (1992) for example). Thus, if we can make an approximation of $\eta(x)^m$, then by taking a summation of those basis, we obtain an approximate of $\mathcal{N}_m(x)$. It is shown by Yarotsky (2017); Schmidt-Hieber (2018) that, for $D \in \mathbb{N}$ and any $\epsilon > 0$, there exists a neural network $\phi_{\mathrm{mult}} \in \Phi(L, W, S, B)$ with $L = \lceil \log_2\left(\frac{3^D}{\epsilon}\right) + 5\rceil\lceil \log_2(D)\rceil$, $W = 6d$, $S = LW^2$ and $B = 1$ such that

$$\sup_{x\in[0,1]^D}\left|\phi_{\mathrm{mult}}(x_1,\ldots,x_D) - \prod_{i=1}^{D}x_i\right| \le \epsilon,$$

and $\phi_{\mathrm{mult}}(0,\ldots,0) = 0$ for $y \in \mathbb{R}^D$ such that $\prod_{j=1}^{D}y_j = 0$. Moreover, for any $M > 0$, we can realize the function $\min\{M, \max\{x, 0\}\}$ by a single-layer neural network $\phi_{(0,M)}(x) := \eta(x) - \eta(x - M)(= \min\{M, \max\{x, 0\}\})$. Thus, for $x \in \mathbb{R}$, it holds that

$$\sup_{x\in[0,M]}\left|\phi_{\mathrm{mult}}(\phi_{(0,1)}(x/M),\ldots,\phi_{(0,1)}(x/M)) - (\phi_{(0,1)}(x/M))^m\right| \le \epsilon.$$

Now, $\mathcal{N}_m(x) = 0$ for $x \notin [0, m+1]$ gives that

$$\mathcal{N}_m(x) = \frac{1}{m!}\sum_{j=0}^{m+1}(-1)^j\binom{m+1}{j}\phi_{(0,m+1-j)}(x-j)^m$$

$$= \frac{1}{m!}\sum_{j=0}^{m+1}(-1)^j\binom{m+1}{j}(m+1)^m\phi_{(0,1-j/(m+1))}((x-j)/(m+1))^m.$$

Therefore, letting

$$f(x) = \frac{1}{m!}\sum_{j=0}^{m+1}(-1)^j(m+1)^m\binom{m+1}{j}\phi_{\mathrm{mult}}\left(\underbrace{\phi_{(0,1-\frac{j}{m+1})}\left(\frac{x-j}{m+1}\right),\ldots,\phi_{(0,1-\frac{j}{m+1})}\left(\frac{x-j}{m+1}\right)}_{m\text{-times}}\right),$$

we have that $f(x) = 0$ for all $x \le 0$ and

$$\sup_{0\le x\le m+1}|\mathcal{N}_m(x) - f(x)| \le \frac{1}{m!}\sum_{j=0}^{m+1}\binom{m+1}{j}(m+1)^m\epsilon \le \frac{(m+1)^m}{\sqrt{2\pi}m^{m+1/2}e^{-m}}2^{m+1}\epsilon$$

$$\le e\frac{(2e)^m}{\sqrt{m}}\epsilon =: \epsilon',$$

where we used $\sum_{j=0}^{m+1}\binom{m+1}{j} = 2^{m+1}$ and Stirling's approximation $m! \ge \sqrt{2\pi}m^{m+1/2}e^{-m}$ in the second inequality. Hence, we also have that, for all $x > m+1$,

$$f(x) = \frac{1}{m!}\sum_{j=0}^{m+1}(-1)^j\binom{m+1}{j}(m+1)^m$$

$$\times \phi_{\mathrm{mult}} \left( \phi_{(0,1-\frac{j}{m+1})} \left( \frac{m+1-j}{m+1} \right), \dots, \phi_{(0,1-\frac{j}{m+1})} \left( \frac{m+1-j}{m+1} \right) \right)$$

$$=: \delta'.$$

It holds that $|\delta'| \leq \epsilon'$. Since it is possible that $\delta' \neq 0$, we modify $f(x)$ so that $f(x) = 0$ outside the interval $[0, m+1]$. Because of this and noting $0 \leq \mathcal{N}_m(x) \leq 1$, we see that $g(x) := \phi_{(0,1)}(f(x) - \frac{\delta'}{m+1}\phi_{(0,m+1)}(x))$ yields

$$\sup_{x \in \mathbb{R}} |\mathcal{N}_m(x) - g(x)| \leq 2\epsilon',$$
$$\sup_{x \in \mathbb{R}} |g(x)| \leq 1, \;\; g(x) = 0 \; (\forall x \notin [0, m+1]).$$

Hence, by applying $\phi_{\mathrm{mult}}$ again, we finally obtain that

$$\sup_{x \in [0,1]^d} |M_{0,0}^d(x) - \phi_{\mathrm{mult}}(g(x_1), \dots, g(x_d))|$$

$$\leq \sup_{x \in [0,1]^d} \left| M_{0,0}^d(x) - \prod_{j=1}^d g(x_j) \right| + \sup_{x \in [0,1]^d} \left| \prod_{j=1}^d g(x_j) - \phi_{\mathrm{mult}}(g(x_1), \dots, g(x_d)) \right|$$

$$\leq 2d\epsilon' + \epsilon.$$

We again applying $\phi_{(0,1)}$, we obtain that $h(x) = \phi_{(0,1)} \circ \phi_{\mathrm{mult}}(g(x_1), \dots, g(x_d))$ satisfies $\|M_{0,0}^d - h\|_{L^\infty(\mathbb{R}^d)} \leq 2d\epsilon' + \epsilon$, $h(x) = 0$ for all $x \notin [0, m+1]^d$, and $\|h\|_\infty \leq 1$. Finally, by carefully checking the network construction, it is shown that $h \in \Phi(L, W, S, B)$ with $L = 3 + 2\lceil \log\left(\frac{3^{d \vee m}}{\epsilon}\right) + 5\rceil\lceil \log_2(d \vee m) \rceil$, $W = 6dm(m+2) + 2d$, $S = LW^2$ and $B = 2(m+1)^m$. Hence, resetting $\epsilon \leftarrow 2d\epsilon' + \epsilon = (1 + 2de\frac{(2e)^m}{\sqrt{m}})\epsilon$, $h$ becomes the desired $\check{M}$. □

## B   PROOF OF PROPOSITION 1

For the order $m \in \mathbb{N}$ of the cardinal B-spline bases, let $J(k) = \{-m, -m+1, \dots, 2^k - 1, 2^k\}^d$ and the quasi-norm of the coefficient $(\alpha_{k,j})_{k,j}$ for $k \in \mathbb{Z}_+$ and $j \in J(k)$ be

$$\|(\alpha_{k,j})_{k,j}\|_{b_{p,q}^s} := \left\{ \sum_{k \in \mathbb{Z}_+} \left[ 2^{k(s-d/p)} \left( \sum_{j \in J(k)} |\alpha_{k,j}|^p \right)^{1/p} \right]^q \right\}^{1/q}.$$

**Lemma 2.** *Under one of the conditions* (4) *in Proposition 1 and the condition* $0 < s < \min(m, m - 1 + 1/p)$ *where* $m \in \mathbb{N}$ *is the order of the cardinal B-spline bases, for any* $f \in B_{p,q}^s(\Omega)$*, there exists* $f_N$ *that satisfies*

$$\|f - f_N\|_{L^r(\Omega)} \lesssim N^{-s/d} \|f\|_{B_{p,q}^s} \tag{9}$$

*for* $N \gg 1$*, and has the following form:*

$$f_N(x) = \sum_{k=0}^K \sum_{j \in J(k)} \alpha_{k,j} M_{k,j}^d(x) + \sum_{k=K+1}^{K^*} \sum_{i=1}^{n_k} \alpha_{k,j_i} M_{k,j_i}^d(x), \tag{10}$$

*where* $(j_i)_{i=1}^{n_k} \subset J(k)$, $K = \lceil C_1 \log(N)/d \rceil$, $K^* = \lceil \log(\lambda N)\nu^{-1} \rceil + K + 1$, $n_k = \lceil \lambda N 2^{-\nu(k-K)} \rceil$ $(k = K+1, \dots, K^*)$ *for* $\delta = d(1/p - 1/r)_+$ *and* $\nu = (s-\delta)/(2\delta)$ *where the real number constants* $C_1 > 0$ *and* $\lambda > 0$ *are chosen to satisfy* $\sum_{k=1}^K (2^k + m)^d + \sum_{k=K+1}^{K^*} n_k \leq N$ *independently to* $N$*. Moreover, we can choose the coefficients* $(\alpha_{k,j})$ *to satisfy*

$$\|(\alpha_{k,j})_{k,j}\|_{b_{p,q}^s} \lesssim \|f\|_{B_{p,q}^s}.$$

*Proof of Lemma 2.* DeVore & Popov (1988) constructed a linear bounded operator $P_k$ having the following form:

$$P_k(f)(x) = \sum_{j \in J(k)} a_{k,j} M_{k,j}^d(x) \tag{11}$$

where $\alpha_{k,j}$ is constructed in a certain way, where for every $f \in L^p([0,1]^d)$ with $0 < p \leq \infty$, it holds

$$\|f - P_k(f)\|_{L^p} \leq C w_{r,p}(f, 2^{-k}). \tag{12}$$

Let

$$p_k(f) := P_k(f) - P_{k-1}(f), \quad P_{-1}(f) = 0.$$

Then, it is shown that for $0 < p, q \leq \infty$ and $0 < s < \min(m, m-1+1/p)$, $f$ belongs to $B_{p,q}^s$ if and only if $f$ can be decomposed into

$$f = \sum_{k=0}^{\infty} p_k(f),$$

with the convergence condition $\|(p_k(f))_{k=0}^{\infty}\|_{b_p^s(L^p)} := (\sum_{k \in \mathbb{Z}_+} (2^{sk} \|p_k\|_{L^p})^q)^{1/q} < \infty$; in particular, $\|f\|_{B_{p,q}^s} \simeq \|(p_k(f))_{k=0}^{\infty}\|_{b_p^s(L^p)}$. Here, each $p_k$ can be expressed as $p_k(x) = \sum_{j \in J(k)} \alpha_{k,j} M_{k,j}^d(x)$ for a coefficient $(\alpha_{k,j})_{k,j}$ which could be different from $(a_{k,j})_{k,j}$ appearing in Eq. (11). Hence, $f \in B_{p,q}^s$ can be decomposed into

$$f = \sum_{k=0}^{\infty} \sum_{j \in J(k)} \alpha_{k,j} M_{k,j}^d(x) \tag{13}$$

with convergence in the sence of $L^p$. Moreover, it is shown that $\|p_k\|_{L^p} \simeq (2^{-kd} \sum_{j \in J(k)} |\alpha_{k,j}|^p)^{1/p}$ and thus

$$\|f\|_{B_{p,q}^s} \simeq \|(\alpha_{k,j})_{k,j}\|_{b_{p,q}^s}. \tag{14}$$

Based on this decomposition, Dũng (2011b) proposed an optimal adaptive recovery method such that the approximator has the form (10) under the conditions for $K, K^*, n_k$ given in the statement and satisfies the approximation accuracy (9). This can be proven by applying the proof of Theorem 3.1 in Dũng (2011b) to the decomposition (13) instead of Eq. (3.8) of that paper. See also Theorem 5.4 of Dũng (2011b). Moreover, the equivalence (14) gives the norm bound of the coefficient $(\alpha_{k,j})$. $\qquad\square$

*Proof of Proposition 1.* Basically, we combine Lemma 1 and Lemma 2. We substitute the approximated cardinal B-spline basis $\check{M}$ into the decomposition of $f_N$ (10). Let the set of indexes $(k,j) \in \mathbb{Z} \times \mathbb{Z}$ that consists $f_N$ given in Eq. (10) be $E_N$; i.e., $f_N = \sum_{(k,j) \in E_N} \alpha_{k,j} M_{k,j}^d$. Accordingly, we set $\check{f} := \sum_{(k,j) \in E_N} \alpha_{k,j} \check{M}_{k,j}^d$. For each $x \in \mathbb{R}^d$, it holds that

$$
\begin{aligned}
|f_N(x) - \check{f}(x)| &\leq \sum_{(k,j) \in E_N} |\alpha_{k,j}| |M_{k,j}^d(x) - \check{M}_{k,j}^d(x)| \\
&\leq \epsilon \sum_{(k,j) \in E_N} |\alpha_{k,j}| \mathbf{1}\{M_{k,j}^d(x) \neq 0\} \\
&\leq \epsilon (m+1)^d (1 + K^*) 2^{K^*(d/p-s)_+} \|f\|_{B_{p,q}^s} \\
&\lesssim \log(N) N^{(\nu^{-1}+d^{-1})(d/p-s)_+} \epsilon \|f\|_{B_{p,q}^s},
\end{aligned}
$$

where we used the definition of $K^*$ in the last inequality. Therefore, for each $f \in U(B_{p,q}^s([0,1]^d))$, it holds that

$$\|f - \check{f}\|_{L^r} \lesssim \|f - f_N\|_{L^r} + \|f_N - \check{f}\|_{L^r} \lesssim \log(N) N^{(\nu^{-1}+d^{-1})(d/p-s)_+} \|f\|_{B_{p,q}^s} \epsilon + N^{-s/d}.$$

By taking $\epsilon$ to satisfy $\log(N) N^{(\nu^{-1}+d^{-1})(d/p-s)_+} \epsilon \leq N^{-s/d}$ (i.e., $\epsilon \leq N^{-s/d-(\nu^{-1}+d^{-1})(d/p-s)_+} \log(N)^{-1}$), then we obtain the approximation error bound.

Next, we bound the magnitude of the coefficients. Each coefficient $\alpha_{j,k}$ satisfies $|\alpha_{j,k}| \lesssim 2^{k(d/p-s)_+} \|f\|_{B_{p,q}^s} \leq 2^{k(d/p-s)_+} \lesssim N^{(\nu^{-1}+d^{-1})(d/p-s)_+}$ for $k \leq K^*$. Finally, the magnitudes of the coefficients hidden in $\check{M}_{k,j}^d$ are evaluated. Remembering that $\check{M}_{k,j}^m(x) = \check{M}(2^k x_1 - j_1, \ldots, 2^k x_d - j_d)$, we see that we just need to bound the quantity $2^k$ $(k \leq K^*)$. However, this is bounded by $2^k \leq N^{\nu^{-1}+d^{-1}}$ for $k \leq K^*$. Hence, we obtain the assertion. $\qquad\square$

## C  PROOF OF THEOREM 1

### C.1  PREPARATION: SPARSE GRID

Here, we give technical details behind the approximation bound. The analysis utilizes the so called *sparse grid* technique Smolyak (1963) which has been developed in the function approximation theory field.

As we have seen in the above, in a typical B-spline approximation scheme, we put the basis functions $M_{k,j}^m(x)$ on a "regular grid" for $k = 1, \ldots, K$ and $(j_1, \ldots, j_d) \in J(k)$, and take its superposition as $f(x) \approx \sum_{k=1,\ldots,K} \sum_{j \in J(k)} \alpha_{k,j} M_{k,j}^m(x)$, which consists of $O(2^{Kd})$ terms (see Eq. (10)). Hence, the number of parameters $O(2^{Kd})$ is affected by the dimensionality $d$ in an exponential order. However, to approximate functions with mixed smoothness, we do not need to put the basis on the whole range of the regular grid. Instead, we just need to put them on a *sparse grid* which is a subset of the regular grid and has much smaller cardinality than the whole set. The approximation algorithm utilizing sparse grid is based on Smolyak's construction (Smolyak, 1963) and its applications to mixed smooth spaces (Dũng, 1990; 1991; 1992; Temlyakov, 1982; 1993a;b). Dũng (2011a) studied an optimal non-adaptive linear sampling recovery method for the mixed smooth Besov space based on the cardinal B-spline bases. We adopt this method, and combining this with the adaptive technique developed in Dũng (2011b), we give the following approximation bound using a non-linear adaptive method to obtain better convergence for the setting $p < r$.

Before we state the theorem, we define an quasi-norm of a set of coefficients $\alpha_{k,j} \in \mathbb{R}$ for $k \in \mathbb{Z}_+^d$ and $j \in J_m^d(k) := \{-m, -m+1, \ldots, 2^{k_1} - 1, 2^{k_1}\} \times \cdots \times \{-m, -m+1, \ldots, 2^{k_d} - 1, 2^{k_d}\}$ as

$$\|(\alpha_{k,j})_{k,j}\|_{mb_{p,q}^s} := \left( \sum_{k \in \mathbb{Z}_+^d} \left[ 2^{(s-1/p)\|k\|_1} \left( \sum_{j \in J_m^d(k)} |\alpha_{k,j}|^p \right)^{1/p} \right]^q \right)^{1/q}.$$

**Theorem 5.** *Suppose that $0 < p, q, r \le \infty$ and $s > (1/p - 1/r)_+$. Assume that the order $m \in \mathbb{N}$ of the cardinal B-spline satisfies $0 < s < \min(m, m - 1 + 1/p)$. Let $\delta = (1/p - 1/r)_+$. Then, for any $f \in MB_{p,q}^s(\Omega)$ and $K > 0$, there exists $R_K(f)$ such that $R_K(f)$ can be represented as*

$$R_K(f)(x) = \sum_{\substack{k \in \mathbb{Z}_+^d: \\ \|k\|_1 \le K}} \sum_{j \in J_m^d(k)} \alpha_{k,j} M_{k,j}^d(x) + \sum_{\substack{k \in \mathbb{Z}_+^d: \\ K < \|k\|_1 \le K^*}} \sum_{i=1}^{n_k} \alpha_{k,j_i^{(k)}} M_{k,j_i^{(k)}}^d(x),$$

*where $K^* = \lceil K(1 + \frac{2\delta}{s-\delta}) \rceil$, $(j_i^{(k)})_{i=1}^{n_k} \subset J_m^d(k)$, and $n_k = \lceil 2^{K - \frac{s-\delta}{2\delta}(\|k\|_1 - K)} \rceil$, and satisfies the following properties:*

(i) *For $p \ge r$,*

$$\|f - R_K(f)\|_r \lesssim 2^{-Ks} D_{K,d}^{(1/\min(r,1) - 1/q)_+} \|f\|_{MB_{p,q}^s}.$$

(ii) *For $p < r$,*

$$\|f - R_K(f)\|_r \lesssim \begin{cases} 2^{-Ks} D_{K,d}^{(1/r - 1/q)_+} \|f\|_{MB_{p,q}^s} & (r < \infty), \\ 2^{-Ks} D_{K,d}^{(1 - 1/q)_+} \|f\|_{MB_{p,q}^s} & (r = \infty). \end{cases}$$

*Moreover, the coefficients $(\alpha_{k,j})_{k,j}$ can be taken to hold $\|(\alpha_{k,j})_{k,j}\|_{mb_{p,q}^s} \lesssim \|f\|_{MB_{p,q}^s}$.*

The proof is given in Appendix C.2. The total number of cardinal B-spline bases consisting of $R_K(f)$ can be evaluated as

$$2^{K+1} \binom{K+d-1}{d-1} + \sum_{k: K < \|k\|_1 \le K^*} n_k$$

$$\lesssim 2^K D_{K,d} + 2^K D_{K^*,d} \lesssim 2^K D_{K,d} \qquad (\because \text{Eq. (17)}).$$

Here, $D_{K,d}$ can be evaluated as

$$D_{K,d} \lesssim K^{d-1} \quad \text{or} \quad D_{K,d} \lesssim d^K.$$

Therefore, the total number of bases can be evaluated as

$$2^K \min\{K^{d-1}, d^K\}$$

which is much smaller than $2^{Kd}$ which is required to approximate functions in the ordinal Besov space (see Lemma 2). In this proposition, $K$ controls the resolution and as $M$ goes to infinity, the approximation error goes to 0 exponentially fast. A remarkable point in the proposition is in the construction of $R_K(f)$ in which the superposition is taken over $\|k\|_1 \leq M$ instead of $\|k\|_\infty \leq K^* = O(K)$. Hence, the number of terms appearing in the summation is at most $O(2^K K^{d-1})$ while the full grid takes $O(2^{Kd})$ terms. This represents how the mixed smoothness is important to ease the curse of dimensionality.

Several aspects of the m-Besov space such as the optimal $N$-term approximation error and Kolmogorov widths have been extensively studied in the literature (see a comprehensive survey (Dũng et al., 2016)). An analogous result is already given by Dũng (2011a) in which $s > 1/p$ is assumed and a linear interpolation method is investigated. However, our result only requires $s > (1/p - 1/q)_+$. This difference comes from a point that our analysis allows nonlinear adaptive interpolation instead of (linear) non-adaptive sampling considered in Dũng (2011a). Because of this, our bound is better than the optimal rate of linear methods (Galeev, 1996; Romanyuk, 2001) and non-adaptive methods (Dũng, 1990; 1991; 1992; Temlyakov, 1982; 1993a;b) especially in the regime of $p < r$ (Dũng (1992) also deals with adaptive methods but does not cover $p < r$ for adaptive method). See Proposition 2 for comparison.

## C.2 PROOF OF THEOREM 5

*Proof of Theorem 5.* For $k = (k_1, \ldots, k_d) \in \mathbb{Z}_+^d$, let $P_{k_i}^{(i)} f(x)$ be the function operating $P_{k_i}$ defined in (11) to $f$ as a function of $x_i$ with other components $x_j$ ($j \neq i$) fixed, and let

$$p_k := \prod_{i=1}^d (P_{k_i}^{(i)} - P_{k_i-1}^{(i)}) f. \tag{15}$$

Then, $p_k$ can be expressed as $p_k(x) = \sum_{j \in J_m^d(k)} \alpha_{k,j} M_{k,j}^d(x)$.

Let $T_{k_i}^{(i)} = I - P_{k_i}^{(i)}$ and $\|f\|_{p,i}$ be the $L^p$-norm of $f$ as a function of $x_i$ with other components $x_j$ ($j \neq i$) fixed (i.e., if $p < \infty$, $\|f\|_{p,i}^p = \int |f(x)|^p \mathrm{d}x_i$), then Eq. (12) gives

$$\|T_{k_i}^{(i)} f\|_{p,i} \lesssim \sup_{|h_i| \leq 2^{-k_i}} \|\Delta_{h_i}^{r,i}(f)\|_{p,i}.$$

Thus, by applying the same argument again, it also holds

$$\|\|T_{k_i}^{(i)} T_{k_j}^{(j)} f\|_{p,i}\|_{p,j} \lesssim \|\sup_{|h_i| \leq 2^{-k_i}} \|\Delta_{h_i}^{r,i}(T_{k_j} f)\|_{p,i}\|_{p,j}$$

$$= \sup_{|h_i| \leq 2^{-k_i}} \|\|T_{k_j} \Delta_{h_i}^{r,i}(f)\|_{p,j}\|_{p,i} \quad (\because \text{the definition of } \Delta_{h_i}^{r,i} \text{ and Fubini's theorem})$$

$$\lesssim \sup_{|h_i| \leq 2^{-k_i}} \sup_{|h_j| \leq 2^{-k_j}} \|\|\Delta_{h_i}^{r,i}(\Delta_{h_j}^{r,j}(f))\|_{p,j}\|_{p,i},$$

for $i \neq j$. Thus, applying the same argument recursively, for $u \subset [d]$, it holds that

$$\left\| \prod_{i \in u} T_{k_i}^{(i)} f \right\|_p \lesssim w_{r,p}^u(f, 2^{-k}),$$

for $k \in \mathbb{Z}_+^d(u)$. Therefore, since $p_k = \prod_{i=1}^d (T_{k_i-1}^{(i)} - T_{k_i}^{(i)}) f = \sum_{u \subset [d]} (-1)^{|u|} \left( \prod_{i \in u} T_{k_i}^{(i)} \prod_{i \notin u} T_{k_i-1}^{(i)} \right) f$, by letting $e = \{i \mid k_i > 0\}$, we have that

$$\|p_k\|_p \lesssim \sum_{u \subset [d]} \left\| \left( \prod_{i \in u} T_{k_i}^{(i)} \prod_{i \notin u} T_{k_i-1}^{(i)} \right) f \right\|_p \lesssim \sum_{u \subset [d]} w_{r,p}^{\hat{u}}(f, 2^{-(k^u)_{\hat{u}}}) \lesssim \sum_{e \subset u} w_{r,p}^u(f, 2^{-k^u})$$

where $k_i^u := k_i$ ($i \in u$) and $k_i^u := k_i - 1$ ($i \notin u$), $\hat{u} = \{i \mid k_i^u \geq 0\}$, and $(k^u)_{\hat{u}}$ is a vector such that $(k^u)_{\hat{u},i} = k_i^u$ for $i \in \hat{u}$ and $(k^u)_{\hat{u},i} = 0$ for $i \notin \hat{u}$. Now let

$$\|(p_k)_k\|_{b_q^\alpha(L^p)} = \left( \sum_{k \in \mathbb{Z}_+^d} \left( 2^{\alpha\|k\|_1} \|p_k\|_{L^p} \right)^q \right)^{1/q}$$

for $p_k \in L^p(\Omega)$ ($k \in \mathbb{Z}_+^d$). Hence, if we set $a_k = \sum_{e \subset u} w_{r,p}^u(f, 2^{-k^u})$ for $k \in \mathbb{Z}_+^d$ and $e = \{i \mid k_i > 0\}$, we have that

$$\|(p_k)_k\|_{b_q^\alpha(L^p)} \lesssim \|(a_k)\|_{b_q^\alpha(L^p)} \simeq \|f\|_{MB_{p,q}^s}.$$

On the other hand, following the same line of Theorem 2.1 (ii) of Dũng (2011a), we also obtain the opposite inequality $\|f\|_{MB_{p,q}^s} \simeq \|(a_k)\|_{b_q^\alpha(L^p)} \lesssim \|(p_k)_k\|_{b_q^\alpha(L^p)}$ (note that the analogous inequality to Lemma 2.3 of Dũng (2011a) also holds in our setting by replacing $q_s$ with $p_s$ and $\omega_r^e(f, 2^{-k})_p$ by $w_{r,p}^e$).

Therefore, $f \in MB_{p,q}^\alpha$ if and only if $(p_k)_{k \in \mathbb{Z}_+^d}$ given by Eq. (15) satisfies $\|(p_k)_k\|_{b_q^\alpha(L^p)} < \infty$ and $f$ can be decomposed into $f = \sum_{k \in \mathbb{Z}_+^d} p_k$ where convergence is in $MB_{p,q}^\alpha$. Moreover, it holds that $\|f\|_{MB_{p,q}^\alpha} \simeq \|(p_k)_k\|_{b_q^\alpha(L^p)}$. This can be shown by Theorem 2.1 of Dũng (2011a). Moreover, by the quasi-norm equivalence $\|p_k\|_p \simeq 2^{-\|k\|_1/p} (\sum_{j \in J_m^d(k)} |\alpha_{k,j}|^p)^{1/p}$, we also have $\|(\alpha_{k,j})_{k,j}\|_{mb_{p,q}^\alpha} \simeq \|f\|_{MB_{p,q}^\alpha}$.

If $p \geq r$, the assertion can be shown in the same manner as Theorem 3.1 of Dũng (2011a).

For the setting of $p < r$, we need to use an adaptive approximation method. In the following, we assume $p < r$. For a given $K$, by choosing $K^*$ appropriately later, we set

$$R_K(f)(x) = \sum_{k \in \mathbb{Z}_+^d : \|k\|_1 \leq K} p_k + \sum_{k \in \mathbb{Z}_+^d : K < \|k\|_1 \leq K^*} G_k(p_k),$$

where $G_k(p_k)$ is given as

$$G_k(p_k) = \sum_{1 \leq i \leq n_k} \alpha_{k,j_i} M_{k,j_i}^d(x)$$

where $(\alpha_{k,j_i})_{i=1}^{|J_m^d(k)|}$ is the sorted coefficients in decreasing order of their absolute value: $|\alpha_{k,j_1}| \geq |\alpha_{k,j_2}| \geq \cdots \geq |\alpha_{k,j_{|J_m^d(k)|}}|$. Then, it holds that

$$\|p_k - G_k(p_k)\|_r \leq \|p_k\|_p 2^{\delta\|k\|_1} n_k^{-\delta},$$

where $\delta := (1/p - 1/r)$ (see the proof of Theorem 3.1 of Dũng (2011b) and Lemma 5.3 of Dũng (2011a)). Moreover, we also have

$$\|p_k\|_r \leq \|p_k\|_p 2^{\delta\|k\|_1}$$

for $k \in \mathbb{Z}_+^d$ with $\|k\|_1 > K^*$.

Here, we define $N$ as

$$N = \lceil \log_2(K) \rceil.$$

Let $\epsilon = (s - \delta)/(2\delta)$, and

$$K^* = \lceil K(1 + 1/\epsilon) \rceil,$$

and $n_k = \lceil 2^{K - \epsilon(\|k\|_1 - K)} \rceil$ for $k \in \mathbb{Z}_+^d$ with $K + 1 \leq \|k\|_1 \leq K^*$.

Then, by Lemma 5.3 of Dũng (2011a), we have that

$$\|f - R_K(f)\|_{L^r}^r \lesssim \sum_{K < \|k\|_1 \leq K^*} \|p_k - G_k(p_k)\|_{L^r}^r + \sum_{K^* < \|k\|_1} \|p_k\|_{L^r}^r$$

$$\lesssim \sum_{K < \|k\|_1 \leq K^*} [\|p_k\|_p 2^{\delta\|k\|_1} n_k^{-\delta}]^r + \sum_{K^* < \|k\|_1} [2^{\delta\|k\|_1} \|p_k\|_{L^p}]^r. \qquad (16)$$

In the following, we require an upper bound of $\binom{k+d-1}{d-1}$. Hence, we evaluate this quantity beforehand. This can be upper bounded by the Stering's formula as

$$\binom{k+d-1}{d-1} \leq \frac{\sqrt{2}e}{2\pi} \underbrace{\left(1+\frac{d-1}{k}\right)^k \left(1+\frac{k}{d-1}\right)^{d-1}}_{=D_{k,d}} \leq D_{k,d}.$$

Let $\xi > 0$ be a positive real number satisfying $1 + \xi \geq K^*/K$. We can see that $\xi$ can be chosen as $\xi = 1/\epsilon + o(1)$. Then, we have that

$$D_{K^*,d} = D_{K,d}\frac{(1+\frac{d-1}{K^*})^{K^*}}{(1+\frac{d-1}{K})^K}\frac{(1+\frac{K^*}{d-1})^{d-1}}{(1+\frac{K}{d-1})^{d-1}} \leq D_{K,d}\frac{(1+\frac{d-1}{K})^{K^*}}{(1+\frac{d-1}{K})^K}\left(\frac{1}{1+\frac{K}{d-1}} + \frac{K^*}{(d-1)(1+\frac{K}{d-1})}\right)^{d-1}$$

$$\leq D_{K,d}\left(1+\frac{d-1}{K}\right)^{K^*-K}\left(\frac{d-1+K^*}{d-1+K}\right)^{d-1} = D_{K,d}\left(1+\frac{d-1}{K}\right)^{\xi K}(1+\xi)^{d-1}$$

$$\leq D_{K,d}e^{(d-1)\xi}(1+\xi)^{d-1} \simeq D_{K,d}. \tag{17}$$

(a) Suppose that $q \leq r$ and $r < \infty$. Then

$$\|f - R_K(f)\|_{L^r}^q = \|f - R_K(f)\|_{L^r}^{r\frac{q}{r}}$$

$$\lesssim \left\{\sum_{K<\|k\|_1\leq K^*}[2^{\delta\|k\|_1}n_k^{-\delta}\|p_k\|_{L^p}]^r + \sum_{K^*<\|k\|_1}[2^{\delta\|k\|_1}\|p_k\|_{L^p}]^r\right\}^{\frac{q}{r}} \quad (\because \text{Eq. (16)})$$

$$\lesssim \sum_{K<\|k\|_1\leq K^*}[2^{\delta\|k\|_1}n_k^{-\delta}\|p_k\|_{L^p}]^q + \sum_{K^*<\|k\|_1}[2^{\delta\|k\|_1}\|p_k\|_{L^p}]^q$$

$$\leq N^{-\delta q}2^{-(s-\delta)Kq}\sum_{K<\|k\|_1\leq K^*}[\underbrace{2^{-(s-\delta-\delta\epsilon)(\|k\|_1-K)}}_{\leq 1}2^{s\|k\|_1}\|p_k\|_{L^p}]^q + 2^{-q(s-\delta)K^*}\sum_{K^*<\|k\|_1}[2^{s\|k\|_1}\|p_k\|_{L^p}]^q$$

$$\lesssim (N^{-\delta}2^{-(s-\delta)K} + 2^{-(s-\delta)K^*})^q\|f\|_{MB_{p,q}^s}^q$$

$$\leq (N^{-s})^q\|f\|_{MB_{p,q}^\alpha}^q.$$

(b) Suppose that $q > r$ and $r < \infty$. Then, letting $\nu = q/r (> 1)$ and $\nu' = 1/(1-1/\nu) = q/(q-r)$ (note that $\frac{1}{\nu} + \frac{1}{\nu'} = 1$), we have

$$\|f - R_K(f)\|_{L^r}^r \lesssim \sum_{K<\|k\|_1\leq K^*}[2^{\delta\|k\|_1}n_k^{-\delta}\|p_k\|_{L^p}]^r + \sum_{K^*<\|k\|_1}[2^{\delta\|k\|_1}\|p_k\|_{L^p}]^r \quad (\because \text{Eq. (16)})$$

$$\leq N^{-\delta r}2^{-(s-\delta)Kr}\sum_{K<\|k\|_1\leq K^*}[2^{-(s-\delta-\delta\epsilon)(\|k\|_1-K)}2^{s\|k\|_1}\|p_k\|_{L^p}]^r + \sum_{K^*<\|k\|_1}[2^{s\|k\|_1}\|p_k\|_{L^p}]^r(2^{-(s-\delta)\|k\|_1})^r$$

$$\leq (N^{-\delta}2^{-(s-\delta)K} + 2^{-(s-\delta)K^*})^r\left\{\sum_{K<\|k\|_1\leq K^*}[2^{-(s-\delta-\delta\epsilon)(\|k\|_1-K)}2^{s\|k\|_1}\|p_k\|_{L^p}]^r\right.$$

$$\left. + \sum_{K^*<\|k\|_1}[2^{s\|k\|_1}\|p_k\|_{L^p}]^r 2^{-(s-\delta)(\|k\|_1-K^*)r}\right\}$$

$$\leq (N^{-\delta}2^{-(s-\delta)K} + 2^{-(s-\delta)K^*})^r\left\{\sum_{K<\|k\|_1\leq K^*}[2^{s\|k\|_1}\|p_k\|_{L^p}]^{r\nu} + \sum_{K^*<\|k\|_1}[2^{s\|k\|_1}\|p_k\|_{L^p}]^{r\nu}\right\}^{1/\nu}$$

$$\times \left\{\sum_{K<\|k\|_1\leq K^*}[2^{-(s-\delta-\delta\epsilon)(\|k\|_1-K)}]^{r\nu'} + \sum_{K^*<\|k\|_1}[2^{-(s-\delta)(\|k\|_1-K^*)}]^{r\nu'}\right\}^{1/\nu'}$$

$$\lesssim (N^{-\delta}2^{-(s-\delta)K} + 2^{-(s-\delta)K^*})^r\|f\|_{MB_{p,q}^s}^r D_{K,d}^{r(1/r-1/q)} \quad (\because \text{Eq. (17)})$$

$$\lesssim (N^{-s}D_{K,d}^{1/r-1/q})^r\|f\|_{MB_{p,q}^s}^r.$$

(c) Suppose that $r = \infty$. Then, similarly to the analysis in (b), we can evaluate

$$\|f - R_K(f)\|_{L^r}$$
$$\lesssim N^{-\delta}2^{-(s-\delta)K} \sum_{K < \|k\|_1 \leq K^*} [2^{-(s-\delta-\delta\epsilon)(\|k\|_1 - K)}2^{s\|k\|_1}\|p_k\|_{L^p}] + \sum_{K^* < \|k\|_1} [2^{s\|k\|_1}\|p_k\|_{L^p}](2^{-(s-\delta)\|k\|_1})$$
$$\lesssim (N^{-\delta}2^{-(s-\delta)K} + 2^{-(s-\delta)K^*})D_{K,d}^{(1-1/q)+}\|f\|_{MB_{p,q}^s}$$
$$\lesssim N^{-s}D_{K,d}^{(1-1/q)+}\|f\|_{MB_{p,q}^s}.$$

$\square$

### C.3 PROOF OF THEOREM 1

Let $\mathbb{Z}_+^d(e) := \{k \in \mathbb{Z}_+^d \mid k_i = 0, i \notin e\}$ and for $k \in \mathbb{Z}_+^d(e)$, we define $2^{-k} := (2^{-k_{i_1}}, \ldots, 2^{-k_{i_{|e|}}}) \in \mathbb{R}_+^{|e|}$ where $\{i_1, \ldots, i_{|e|}\} = e$. By defining $\|(g_k)_k\|_{b_q^{s,e}} := \left(\sum_{k \in \mathbb{Z}_+^d(e)} (2^{s\|k\|_1}|g_k|)^q\right)^{1/q}$ for a sequence $(g_k)_{k \in \mathbb{Z}_+^d(e)}$, then it holds that

$$|f|_{MB_{p,q}^{s,e}} \simeq \sum_{e \subset \{1,\ldots,d\}} \|(w_{r,p}^e(f, 2^{-k}))_k\|_{b_q^{s,e}}.$$

Then, we can prove Theorem 1 based on Theorem 5 as follows.

*Proof of Theorem 1.* The result is immediately follows from Theorem 5. Let the set of indexes of $(k,j)$ consisting of $R_K$ be $E_K$: $R_K(f) = \sum_{(k,j) \in E_K} \alpha_{k,j}M_{k,j}^d(x)$. As in the proof of Proposition 1, we approximate $R_K(f)$ by a neural network given as

$$\check{f}(x) = \sum_{(k,j) \in E_K} \alpha_{k,j}\check{M}_{k,j}^d(x).$$

Each coefficient $\alpha_{j,k}$ satisfies $|\alpha_{j,k}| \lesssim 2^{\|k\|_1(1/p-s)+}\|f\|_{MB_{p,q}^s} \lesssim 2^{K^*(1/p-s)+}$. The difference between $R_K(f)$ and $\check{f}$ can be evaluated as

$$|R_K(f) - \check{f}(x)| \leq \sum_{(k,j) \in E_K} |\alpha_{k,j}||M_{k,j}^d(x) - \check{M}_{k,j}^d(x)|$$
$$\leq \epsilon \sum_{(k,j) \in E_K} |\alpha_{k,j}|\mathbf{1}\{M_{k,j}^d(x) \neq 0\}$$
$$\lesssim \epsilon(m+1)^d(1 + K^*)D_{K^*,d}2^{K^*(1/p-s)+}\|f\|_{MB_{p,q}^s}.$$

Therefore, by taking $\epsilon$ so that $\epsilon(m+1)^d(1 + K^*)D_{K^*,d}2^{K^*(1/p-s)+} \leq 2^{-Ks}$ is satisfied, it holds that

$$|R_K(f) - \check{f}(x)| \lesssim 2^{-Ks}.$$

By the inequality $D_{K^*,d} \leq e^{K^*+d-1}$, it suffices to let $\epsilon \leq \frac{e^{-K^*(s+(1/p-s)_++1)}}{[e(m+1)]^d(1+K^*)}$. The cardinality of $E(K)$ is bounded as

$$\sum_{\kappa=0,\ldots,K} 2^\kappa \binom{\kappa+d-1}{d-1} + \sum_{k:K < \|k\|_1 \leq K^*} n_k$$
$$\leq 2^{K+1}\binom{K+d-1}{d-1} + \sum_{K < \kappa \leq K^*} 2^{K - \frac{s-\delta}{2\delta}(\kappa-K)}\binom{\kappa+d-1}{d-1}$$
$$\leq 2^{K+1}D_{K,d} + 2^K(1 - 2^{-\frac{s-\delta}{2\delta}})^{-1}D_{K^*,d} \leq (2 + (1 - 2^{-\frac{s-\delta}{2\delta}})^{-1})2^K D_{K^*,d} = N.$$

Since each unit $\check{M}_{k,j}^d$ requires width $W_0$, the whole width becomes $W = NW_0$. The number of nonzero parameters to construct $\check{M}_{k,j}^d$ is bounded by $S = (L-1)W_0^2N + N$.

Finally, the magnitudes of the coefficients hidden in $\check{M}_{k,j}^d$ are evaluated. Remembering that $\check{M}_{k,j}^d(x) = \check{M}(2^{k_1}x_1 - j_1, \ldots, 2^{k_d}x_d - j_d)$, here maximum of $2^{k_j}$ is bounded by $2^{K^*} \lesssim N^{(1+1/\nu)}$. Hence, we obtain the assertion. Similarly, it holds that $|\alpha_{j,k}| \lesssim N^{(1+1/\nu)\{1\vee(1/p-s)_+\}}$. $\square$

## D  PROOFS OF THEOREMS 2 AND 3

*Proof of Theorem 2.* We use Proposition 4. We just need to evaluate the covering number of $\hat{\mathcal{F}} = \{\bar{f} \mid f \in \Psi(L, W, S, B)\}$ for $(L, W, S, B)$ given in Theorem 1 where $\bar{f}$ is the clipped function for a given $f$. Note that the covering number of $\hat{\mathcal{F}}$ is not larger than that of $\Psi(L, W, S, B)$. Hence, it is sufficient to evaluate that of $\Psi(L, W, S, B)$. From Lemma 3, the covering number is obtained as

$$\log N(\delta, \hat{\mathcal{F}}, \|\cdot\|_\infty) \lesssim N \log(N)[\log(N)^2 + \log(\delta^{-1})].$$

From Proposition 1, it holds that

$$\|f^\circ - R_K(f^\circ)\|_2 \lesssim N^{-s/d}.$$

Note that

$$\|f - f^\circ\|_{L^2(P_X)}^2 \lesssim \|f - f^\circ\|_2^2.$$

for any $f : [0,1]^d \to \mathbb{R}$ because $p(x) \leq R$. Therefore, by applying Proposition 4 with $\delta = 1/n$, we have that

$$\mathrm{E}_{D_n}[\|\hat{f} - f^\circ\|_{L^2(P_X)}^2] \lesssim N^{-2s/d} + \frac{N \log(N)(\log(N)^2 + \log(n))}{n} + \frac{1}{n}. \tag{18}$$

Here, the right hand side is minimized by setting $N \asymp n^{\frac{d}{2s+d}}$ up to $\log(n)^3$-order, and then have an upper bound of the RHS as

$$n^{-\frac{2s}{2s+d}} \log(n)^3.$$

This gives the assertion. $\square$

*Proof of Theorem 3.* The proof follows the almost same line as the proof of Theorem 2. By noting $S = O(2^K D_{K,d})$, $L = O(K)$ and $W = O(2^K D_{K,d})$, Lemma 3 gives an upper bound of the covering number as

$$\log N(\delta, \hat{\mathcal{F}}, \|\cdot\|_\infty) \lesssim 2^K D_{K,d}[K^2 \log(2^K D_{K,d}) + \log(\delta^{-1})] \lesssim 2^K D_{K,d}(K^3 + \log(1/\delta)).$$

Letting $r = 2$, we have that

$$\|f^\circ - R_K(f^\circ)\|_2 \lesssim 2^{-sK} D_{K,d}^u$$

where $u = (1 - 1/q)_+$ for $p \geq 2$ and $u = (1/2 - 1/q)_+$ for $p < 2$.

Then, by noting that

$$\|f - f^\circ\|_{L^2(P_X)}^2 \lesssim \|f - f^\circ\|_2^2,$$

for any $f : [0,1]^d \to \mathbb{R}$, and by applying Proposition 4 with $\delta = 1/n$, we have that

$$\mathrm{E}_{D_n}[\|\hat{f} - f^\circ\|_{L^2(P_X)}^2] \lesssim 2^{-2sK} D_{K,d}^{2u} + \frac{2^K D_{K,d}(K^3 + \log(\delta^{-1}))}{n} + \frac{1}{n}. \tag{19}$$

Here, we use the following evaluations for $D_{K,d}$: (a) $D_{K,d} \lesssim K^{d-1}$, and (b) $D_{K,d} \lesssim [e(1 + \frac{d}{K})]^K$.

(a) For the evaluation, $D_{K,d} \lesssim K^{d-1}$, we have an upper bound of the right hand side of Eq. (19) as

$$2^{-2sK} K^{2u(d-1)} + \frac{2^K K^{d-1}(K^3 + \log(n))}{n},$$

which is minimized by setting $K = \lceil \frac{1}{1+2s} \log_2(n) + \frac{(2u-1)(d-1)}{1+2s} \log_2 \log(n) \rceil$ up to $\log \log(n)$-order. In this situation, we have the generalization error bound as

$$n^{-\frac{2s}{2s+1}} \log(n)^{\frac{2(d-1)(u+s)}{1+2s}} \log(n)^3.$$

(b) For the evaluation, $D_{K,d} \lesssim [e(1 + \frac{d}{K})]^K \leq e^K e^d$, Eq. (19) gives an upper bound of

$$2^{-2sK} e^{2uK} + \frac{2^K e^K (K^2 + \log(n))}{n}.$$

Then, the right hand side is minimized by $K = \lceil \frac{1}{1+2s+(1-2u)\log_2(e)} \log_2(n) \rceil$. Then, we have that

$$n^{-\frac{2s-2u\log_2(e)}{1+2s+(1-2u)\log_2(e)}} \log(n)^2.$$

This gives the assertion. $\qquad\qquad\qquad\qquad\qquad\qquad\qquad\qquad\qquad\qquad\qquad\qquad\quad\square$

## E  MINIMAX OPTIMALITY

*Proof of Theorem 4.*  First note that since $P_X$ is the uniform distribution, it holds that $\| \cdot \|_{L^2(P_X)} = \| \cdot \|_{L^2([0,1]^d)}$. The $\epsilon$-covering number $\mathcal{N}(\epsilon, \mathcal{G}, L^2(P_X))$ with respect to $L^2(P_X)$ for a function class $\mathcal{G}$ is the minimal number of balls with radius $\epsilon$ measured by $L^2(P_X)$-norm needed to cover the set $\mathcal{G}$ (van der Vaart & Wellner, 1996). The $\delta$-packing number $\mathcal{M}(\delta, \mathcal{G}, L^2(P_X))$ of a function class $\mathcal{G}$ with respect to $L^2(P_X)$ norm is the largest number of functions $\{f_1, \ldots, f_{\mathcal{M}}\} \subseteq \mathcal{G}$ such that $\|f_i - f_j\|_{L^2(P_X)} \geq \delta$ for all $i \neq j$. It is easily checked that

$$\mathcal{N}(\delta/2, \mathcal{G}, L^2(P_X)) \leq \mathcal{M}(\delta, \mathcal{G}, L^2(P_X)) \leq \mathcal{N}(\delta, \mathcal{G}, L^2(P_X)). \tag{20}$$

For a given $\delta_n > 0$ and $\varepsilon_n > 0$, let $Q$ be the $\delta_n$ packing number $\mathcal{M}(\delta_n, U(MB_{p,q}^s), L^2(P_X))$ of $U(MB_{p,q}^s)$ and $N$ be the $\varepsilon_n$ covering number of that. Raskutti et al. (2012) utilized the techniques developed by Yang & Barron (1999) to show the following inequality in their proof of Theorem 2(b):

$$\inf_{\widehat{f}} \sup_{f^* \in U(MB_{p,q}^s)} \mathrm{E}_{D_n}[\|\widehat{f} - f^*\|_{L^2(P_X)}^2] \geq \inf_{\widehat{f}} \sup_{f^* \in U(MB_{p,q}^s)} \frac{\delta_n^2}{2} P[\|\widehat{f} - f^*\|_{L^2(P_X)}^2 \geq \delta_n^2/2]$$

$$\geq \frac{\delta_n^2}{2} \left(1 - \frac{\log(N) + \frac{n}{2\sigma^2}\varepsilon_n^2 + \log(2)}{\log(Q)}\right).$$

Thus by taking $\delta_n$ and $\varepsilon_n$ to satisfy

$$\frac{n}{2\sigma^2}\varepsilon_n^2 \leq \log(N), \tag{21}$$

$$8 \log(N) \leq \log(Q), \tag{22}$$

$$4 \log(2) \leq \log(Q), \tag{23}$$

the minimax rate is lower bounded by $\frac{\delta_n^2}{4}$. This can be achieved by properly setting $\varepsilon_n \simeq \delta_n$. Now, for given $N$ with respect to $\delta_n > 0$, $M = \log(N)$ satisfies

$$\delta_n \gtrsim M^{-s} \log(M)^{(d-1)(s+1/2-1/q)_+}$$

(Theorem 6.24 of Dũng et al. (2016)). Hence, it suffices to take

$$M \simeq n^{\frac{1}{2s+1}} \log(n)^{\frac{2(d-1)(s+1/2-1/q)_+}{2s+1}}, \tag{24}$$

$$\varepsilon_n \simeq \delta_n \simeq n^{-\frac{2s}{2s+1}} \log(n)^{\frac{2(d-1)(s+1/2-1/q)_+}{2s+1}}, \tag{25}$$

which gives the assertion.

$$\qquad\qquad\qquad\qquad\qquad\qquad\qquad\qquad\qquad\qquad\qquad\qquad\qquad\qquad\qquad\quad\square$$

## F  AUXILIARY LEMMAS

Let the $\epsilon$-covering number with respect to $L^2(P_X)$ for a function class $\mathcal{G}$ be $\mathcal{N}(\epsilon, \mathcal{G}, L^2(P_X))$ as defined in the proof of Theorem 4.

**Proposition 4** (Schmidt-Hieber (2018)). *Let $\mathcal{F}$ be a set of functions. Let $\widehat{f}$ be the least squares estimator in $\mathcal{F}$:*

$$\widehat{f} = \operatorname*{argmin}_{f \in \mathcal{F}} \sum_{i=1}^{n} (y_i - f(x_i))^2.$$

*Assume that $\|f^\circ\|_\infty \leq F$ and all $f \in \mathcal{F}$ satisfies $\|f\|_\infty \leq F$ for some $F \geq 1$. If $\delta > 0$ satisfies $\mathcal{N}(\delta, \mathcal{F}, \|\cdot\|_\infty) \geq 3$, then it holds that*

$$\mathrm{E}_{D_n}[\|\widehat{f} - f^\circ\|_{L^2(P_X)}^2] \leq C \left[ \inf_{f \in \mathcal{F}} \|f - f^\circ\|_{L^2(P_X)}^2 + (F^2 + \sigma^2) \frac{\log \mathcal{N}(\delta, \mathcal{F}, \|\cdot\|_\infty)}{n} + \delta(F + \sigma) \right],$$

*where $C$ is a universal constant.*

*Proof of Proposition 4.* This is almost direct consequence of Lemma 10 of Schmidt-Hieber (2018)[5]. The only difference is the assumption of $\|f\|_\infty \leq F$ for $f \in \mathcal{F}$ and $f = f^\circ$ while Lemma 10 of Schmidt-Hieber (2018) assumed $0 \leq f(x) \leq F'$ for $F' > 1$. However, this can be easily fixed by shifting the function value by $+F$ then the range of $f$ is modified to $[0, 2F]$. Then, our situation is reduced to that of Lemma 10 of Schmidt-Hieber (2018) by substituting $F' \leftarrow 2F$. □

**Lemma 3** (Covering number evaluation). *The covering number of $\Phi(L, W, S, B)$ can be bounded by*

$$\log \mathcal{N}(\delta, \Phi(L, W, S, B), \|\cdot\|_\infty) \leq S \log(\delta^{-1} L (B \vee 1)^{L-1} (W+1)^{2L})$$
$$\leq 2SL \log((B \vee 1)(W+1)) + S \log(\delta^{-1} L).$$

*Proof of Lemma 3.* Given a network $f \in \Phi(L, W, S, B)$ expressed as

$$f(x) = (\mathcal{W}^{(L)} \eta(\cdot) + b^{(L)}) \circ \cdots \circ (\mathcal{W}^{(1)} x + b^{(1)}),$$

let

$$\mathcal{A}_k(f)(x) = \eta \circ (\mathcal{W}^{(k-1)} \eta(\cdot) + b^{(k-1)}) \circ \cdots \circ (\mathcal{W}^{(1)} x + b^{(1)}),$$

and

$$\mathcal{B}_k(f)(x) = (\mathcal{W}^{(L)} \eta(\cdot) + b^{(L)}) \circ \cdots \circ (\mathcal{W}^{(k)} \eta(x) + b^{(k)}),$$

for $k = 2, \ldots, L$. Corresponding to the last and first layer, we define $\mathcal{B}_{L+1}(f)(x) = x$ and $\mathcal{A}_1(f)(x) = x$. Then, it is easy to see that $f(x) = \mathcal{B}_{k+1}(f) \circ (\mathcal{W}^{(k)} \cdot + b^{(k)}) \circ \mathcal{A}_k(f)(x)$. Now, suppose that a pair of different two networks $f, g \in \Phi(L, W, S, B)$ given by

$$f(x) = (\mathcal{W}^{(L)} \eta(\cdot) + b^{(L)}) \circ \cdots \circ (\mathcal{W}^{(1)} x + b^{(1)}), \ g(x) = (\mathcal{W}^{(L)'} \eta(\cdot) + b^{(L)'}) \circ \cdots \circ (\mathcal{W}^{(1)'} x + b^{(1)'}),$$

has a parameters with distance $\delta$: $\|\mathcal{W}^{(\ell)} - \mathcal{W}^{(\ell)'}\|_\infty \leq \delta$ and $\|b^{(\ell)} - b^{(\ell)'}\|_\infty \leq \delta$. Now, not that $\|\mathcal{A}_k(f)\|_\infty \leq \max_j \|\mathcal{W}_{j,:}^{(k-1)}\|_1 \|\mathcal{A}_{k-1}(f)\|_\infty + \|b^{(k-1)}\|_\infty \leq WB \|\mathcal{A}_{k-1}(f)\|_\infty + B \leq (B \vee 1)(W+1) \|\mathcal{A}_{k-1}(f)\|_\infty \leq (B \vee 1)^{k-1} (W+1)^{k-1}$, and similarly the Lipshitz continuity of $\mathcal{B}_k(f)$ with respect to $\|\cdot\|_\infty$-norm is bounded as $(BW)^{L-k+1}$. Then, it holds that

$$|f(x) - g(x)|$$
$$= \left| \sum_{k=1}^{L} \mathcal{B}_{k+1}(g) \circ (\mathcal{W}^{(k)} \cdot + b^{(k)}) \circ \mathcal{A}_k(f)(x) - \mathcal{B}_{k+1}(g) \circ (\mathcal{W}^{(k)'} \cdot + b^{(k)'}) \circ \mathcal{A}_k(f)(x) \right|$$
$$\leq \sum_{k=1}^{L} (BW)^{L-k} \|(\mathcal{W}^{(k)} \cdot + b^{(k)}) \circ \mathcal{A}_k(f)(x) - (\mathcal{W}^{(k)'} \cdot + b^{(k)'}) \circ \mathcal{A}_k(f)(x)\|_\infty$$
$$\leq \sum_{k=1}^{L} (BW)^{L-k} \delta [W(B \vee 1)^{k-1} (W+1)^{k-1} + 1]$$

---

[5]We noticed that there exit some technical flaws in the proof of the lemma, e.g., an incorrect application of the uniform bound to derive the risk of an estimator. However, these flaws can be fixed and the statement itself holds with a slight modification.

$$\leq \sum_{k=1}^{L} (BW)^{L-k} \delta (B \vee 1)^{k-1}(W+1)^k \leq \delta L(B \vee 1)^{L-1}(W+1)^L.$$

Thus, for a fixed sparsity pattern (the locations of non-zero parameters), the covering number is bounded by $\left(\delta/[L(B \vee 1)^{L-1}(W+1)^L]\right)^{-S}$. There are the number of configurations of the sparsity pattern is bounded by $\binom{(W+1)^L}{S} \leq (W+1)^{LS}$. Thus, the covering number of the whole space $\Phi$ is bounded as

$$(W+1)^{LS} \left\{\delta/[L(B \vee 1)^{L-1}(W+1)^L]\right\}^{-S} = [\delta^{-1}L(B \vee 1)^{L-1}(W+1)^{2L}]^S,$$

which gives the assertion.

$\square$

