# OpenReview forum: "Adaptivity of deep ReLU network for learning in Besov and mixed smooth Besov spaces: optimal rate and curse of dimensionality"
_ICLR.cc/2019/Conference_

### Official Review · AnonReviewer3 · 2018-11-02
**Are piecewise linear estimators really minimax optimal for piecewise polynomial signals?**

**Rating:** 6
**Confidence:** 2

**Review:**

This paper describes approximation and estimation error bounds for functions in Besov spaces using estimators corresponding to deep ReLU networks. The general idea of connecting network parameters such as depth, width, and sparsity to classical function spaces is interesting and could lead to novel insights into how and why these networks work and under what settings. The authors carefully define Besov spaces and related literature, and overall the paper is clearly written.

Despite these strengths, I'm left with several questions about the results. The most critical is this: piecewise polynomials are members of the Besov spaces of interest, and ReLU networks produce piecewise linear functions. How can piecewise linear approximations of piecewise polynomial functions lead to minimax optimal rates? The authors' analysis is based on cardinal B-spline approximations, which generally makes sense, but it seems like you would need more terms in a superposition of B-splines of order 2 (piecewise linear) than higher orders to approximate a piecewise polynomial to within a given accuracy. The larger number of terms should lead to worse estimation errors, which is contrary to the main result of the paper. I don't see how to reconcile these ideas.

A second question is about the context of some broad claims, such as that the rates achieved by neural networks cannot be attained by any linear or nonadaptive method. Regarding linear methods, I agree with the author, but I feel like this aspect is given undue emphasis. The key paper cited for rates for linear methods is the Donoho and Johnstone Wavelet Shrinkage paper, in which they clearly show that nonlinear, nonadaptive wavelet shrinkage estimators do indeed achieve minimax rates (within a log factor) for Besov spaces. Given this, how should I interpret claims like "any linear/non-linear approximator
with fixed N -bases does not achieve the approximation error ... in some parameter settings such as 0 < p < 2 < r "?
Wavelets provide a fixed N-basis and achieve optimal rates for Besov spaces. Is the constraint on p and r a setting in which wavelet optimality breaks down? If not, then I don't think the claim is correct. If so, then it would be helpful to understand how relevant this regime for p and r is to practical settings (as opposed to being an edge case).

The work on mixed Besov spaces (e.g. tensor product space of 1-d Besov spaces) is a fine result but not surprising.

A minor note: some of the references are strange, like citing a 2015 paper for minimax rates for Besov spaces that have been known for far longer or a 2003 paper that describes interpolation spaces that were beautifully described in DeVore '98. It would be appropriate to cite these earlier sources.

---

> ### Author Response · Authors · 2018-11-14
> **Reply from authros**
>
> We would appreciate your insightful comments.
>
> (1)
> Q: The most critical is this: piecewise polynomials are members of the Besov spaces of interest, and ReLU networks produce piecewise linear functions. How can piecewise linear approximations of piecewise polynomial functions lead to minimax optimal rates?
>
> A: Thank you for raising a concern about an important point. As you concern, a piecewise linear approximation does "not" achieve the minimax optimal rate. This is because we need large number of linear pieces to approximate smooth functions. Hence, as long as we use a shallow network, we can not achieve the minimax rate with ReLU activation. However, the situation is very different if we use a deep neural network. Actually, the number of pieces "exponentially" grows up as the depth increases, and this property enables us to approximate the higher order B-spline bases by ReLU-DNN with a log(n)-order size. This is the main reason why we can achieve the minimax rate (up to log(n)-term) by the ReLU-DNN.
>
> (2)
> Q: A second question is ... how should I interpret claims like "any linear/non-linear approximator
> with fixed N -bases does not achieve the approximation error ... in some parameter settings such as 0 < p < 2 < r "?
> Wavelets provide a fixed N-basis and achieve optimal rates for Besov spaces.
>
> A:
> Indeed, the Donoho-Johnstone's wavelet shrinkage estimator achieves the minimax optimal rate in terms of the estimation error. Here, we would like to emphasize that the approximation error analysis and estimation error analysis are separated (although they are closely connected): the approximation error is evaluated by the number of bases and the estimation error is evaluated by the sample size. As for the approximation error analysis, the shrinkage estimator should prepare a huge number of bases beforehand which could be much larger than the number of non-zero parameters selected by the shrinkage estimator. In that sense, there is no contradiction about the approximation error analysis because the Kolmogorov width states only about the total number of bases which should be prepared beforehand. On the other hand, a remarkable property of the shrinkage estimator is that it appropriately selects a small number of subsets of the bases in an adaptive way (in this sense, the wavelet shrinkage is an adaptive method). Consequently, it achieves the minimax optimal estimation error rate although the whole number of parameters is still large compared with the selected non-zero components. This adaptivity highly relies on the non-linearity of the soft thresholding operator.
> On the other hand, the deep neural network directly constructs the necessary number of bases for each function in the Besov space. This contrasts the shrinkage estimator; that is, the shrinkage estimator prepare a large number of bases first and then selects a small subset of them (which leads to the minimax optimal estimation error), on the other hand, deep learning directly generates the required bases.
> This difference would be analogous to the relation between a sparse estimator and a low rank matrix estimator.
>
> The difference between the linear estimator and the nonlinear estimator occurs when p < 2 = r (Proposition 3). Interestingly, this is the regime where the approximation errors are different between adaptive methods and non-adaptive ones (see Eq.(5)). In this setting (p < 2 = r), functions is the Besov space can has high spatially inhomogeneous smoothness which is hard to be captured by a linear method.
>
> (3)
> Q: A minor note: some of the references are strange
>
> A:
> Thank you for your informative suggestion. The citation [Gine & Nickl, 2015] for the minimax optimal rate on a Besov spaces is a comprehensive text book that was not intended to be the original paper but just a nice reference to overview the literature. The reference [Adams & Fournier, 2003] for the interpolation space is also referred as a text book describing overview of the literature and several related topics in details. But, as you pointed out, it is more appropriate to cite original papers. We have cited [Kerkyacharian & Picard, 1992; Donoho et al., 1996] for the minimax estimation rate in a Besov pace, and cited [DeVore, 1998] for the interpolation space characterization of a Besov space.
>
>
> G. Kerkyacharian and D. Picard. Density estimation in besov spaces. Statistics & Probability
> Letters, 13:15--24, 1992.
>
> D. L. Donoho, I. M. Johnstone, G. Kerkyacharian, and Dominique Picard. Density esti-
> mation by wavelet thresholding. The Annals of Statistics, 24(2):508--539, 1996.
>
> D. L Donoho, I. M. Johnstone, G. Kerkyacharian and Dominique Picard. Minimax estimation via wavelet shrinkage. The Annals of Statistics, 26(3):879--921, 1998.
>
> R. DeVore. Nonlinear approximation. Acta numerica, 7:51--150, 1998.

---

### Official Review · AnonReviewer1 · 2018-11-03
**Paper that establishes minimax optimal rates for deep network models over Besov spaces**

**Rating:** 6
**Confidence:** 2

**Review:**

This paper makes two contributions:
* First, the authors show that function approximation over Besov spaces for the family of deep ReLU networks of a given architecture provide better approximation rates than linear models with the same number of parameters.
* Second, for this family and this function class they show minimax optimal sample complexity rates for generalization error incurred by optimizing the empirical squared error loss.

Clarity: Very dense; could benefit from considerably more exposition.

Originality: afaik original. Techniques seem to be inspired by a recent paper by Montanelli and Du (2017).

Significance: unclear.

Pros and cons:
This is a theory paper that focuses solely on approximation properties of deep networks. Since there is no discussion of any learning procedure involved, I would suggest that the use of the phrase "deep learning" throughout the paper be revised.

The paper is dense and somewhat inaccessible. Presentation could be improved by adding more exposition and comparisons with existing results.

The generalization bounds in Section 4 are given for an ideal estimator which is probably impossible to compute.

---

> ### Author Response · Authors · 2018-11-14
> **Reply from authors**
>
> We would appreciate your detailed feedback on our manuscript.
>
> (1)
> Q:
> Since there is no discussion of any learning procedure involved, I would suggest that the use of the phrase "deep learning" throughout the paper be revised.
>
> A:
> Thank you for your suggestion. As you pointed out, using another terminology such as a "regularized empirical risk minimizer" might be more specific instead of "deep learning." However, the purpose of this paper is to show the superiority and limitation of deep neural network approaches by investigating its best achievable performance. Hence, we would prefer the terminology "deep learning" to indicate the regularized empirical risk minimization over the deep neural network model. We have added a footnote in page 2 which clarifies what kind of estimator is considered throughout the paper.
>
> (2)
> Q:
> Very dense; could benefit from considerably more exposition.
> The paper is dense and somewhat inaccessible. Presentation could be improved by adding more exposition and comparisons with existing results.
>
> A:
> Due to the space limitations, we should have omitted some detailed explanations, though we did our best to include necessary amount of expositions and comparisons. But, as you pointed out, more explanations would help readability. Hence, we have added some text explanations in page 4 for the definition of the m-Besov space. We also added some explanations for the meaning of the approximation error rate and its relation to depth, width and sparsity after Proposition 1 and Theorem 1.
>
> (3)
> Q: The generalization bounds in Section 4 are given for an ideal estimator which is probably impossible to compute.
>
> A:
> We believe that it is informative to investigate how well deep learning can potentially achieve even in the ideal case (of course, without any cheating) because we cannot say anything about the limitation of deep learning approaches without this kind of investigation. Actually, we think this type of analysis is becoming popular in the statistics community. Moreover, recent intensive studies about convergence properties of SGD for deep learning implies that it is not so much vacuous to assume we can achieve the global optimal solution with a good generalization guarantee. In addition, we can also involve the optimization error in our estimation error bound, but we have omitted that for better readability.

---

### Official Review · AnonReviewer2 · 2018-11-05
**Nice and Relevant Results**

**Rating:** 8
**Confidence:** 2

**Review:**

Summary:
========
The paper presents rates of convergence for estimating nonparametric functions in Besov
spaces using deep NNs with ReLu activations. The authors show that deep Relu networks,
unlike linear smoothers, can achieve minimax optimality. Moreover, they show that in a
restricted class of functions called mixed Besov spaces, there is significantly milder
dependence on dimensionality. Even more interestingly, the Relu network is able to
adapt to the smoothness of the problem.

While I am not too well versed on the background material, my educated guess is that the
results are interesting and relevant, and that the analysis is technically sound.



Detailed Comments:
==================


My main criticism is that the total rate of convergence (estimation error + approximation
error) has not been presented in a transparent way. The estimation error takes the form
of many similar results in nonparametric statistics, but the approximation error is
given in terms of the parameters of the network, which depends opaquely on the dimension
and other smoothness parameters. It is not clear which of these terms dominate, and
consequently, how the parameters W, L etc. should be chosen so as to balance them.


While the mixed Besov spaces enables better bounds, the condition appears quite strong.
In fact, the lower bound is better than for traditional Holder/Sobolev classes. Can you
please comment on how th m-Besov space compares to Holder/Sobolev classes? Also, can
you similiarly define mixed Holder/Sobolev spaces where traditional linear smoothers
might achieve minimax optimal results?


Minor:
- Defn of Holder class: you can make this hold for integral beta if you define m to be
the smallest integer less than beta (e.g. beta=7, m=6). Imo, this is standard in most
texts I have seen.
- The authors claim that the approximation error does not depend on the dimensionality
  needs clarification, since N clearly depends on the dimension. If I understand
  correctly, the approximation error is in fact becoming smaller with d for m-Besov
  spaces (since N is increasing with d), and what the authors meant was that the
  exponential dependnence on d has now been eliminated. Is this correct?

Other
- On page 4, what does the curly arrow notation mean?
- Given the technical nature of the paper, the authors have done a good job with the
  presentation. However, in some places the discussion is very equation driven. For e.g.
  in the 2nd half of page 4, it might help to explain many of the quantities presented in
  plain words.



Confidence: I am reasonably familiar with the nonparametric regression literature, but
not very versed on the deep learning theory literature. I did not read the proofs in
detail.

---

> ### Author Response · Authors · 2018-11-14
> **Reply from authors (1/2)**
>
> Thank you for your suggestive comments. We have revised our paper according to your comments, though unfortunately some of them could not be addressed due to the lack of space.
>
> (1)
> Q: My main criticism is that the total rate of convergence (estimation error + approximation error) has not been presented in a transparent way. ... how the parameters W, L etc. should be chosen so as to balance them.
>
> A:
> We have presented the approximation error bound in a concrete way to minimize misunderstandings, which would have made the presentation a bit opaque instead. The error bound O(N^{-s/d}) is rather typical notation in the approximation theory. Since we think the parameters s,p,q,d as constants, the approximation error in Proposition 1 can be written as R_r = O(N^{-s/d}) under the conditions L=O(log(N)) (depth), W = O(N) (width) and S = O(N log(N)) (sparsity) for an integer N. Roughly speaking, N corresponds the number of parameters, S, upto log(N) order. Thus the convergence rate is written as a function of the number of parameters under an appropriate choice of depth L and width W. We can see that the convergence rate of the error is completely controlled by the smoothness s and the dimensionality d against the number of parameters S. On the other hand, as for the m-Besov case, the approximation error is evaluated as O(N^{-s} \log^{s(d-1)}(N)) for L=O(log(N)), W = O(N) and S = O(N log(N)) for an integer N. Here we again observe that the convergence rate is controlled by the smoothness s and the dimensionality d. We think these representations are more transparent. We have added sentences to explain these relations just after Proposition 1 and Theorem 1.
>
> (2)
> Q: While the mixed Besov spaces enables better bounds, the condition appears quite strong. In fact, the lower bound is better than for traditional Holder/Sobolev classes. Can you please comment on how th m-Besov space compares to Holder/Sobolev classes? Also, can you similarly define mixed Holder/Sobolev spaces where traditional linear smoothers might achieve minimax optimal results?
>
> A:
> Yes, the condition for the mixed Besov space is much stronger than the ordinary Besov space. Yes, we can define mixed smooth Holder/Sobolev space. They are defined just by setting p=q=infty or p=q=2. Hence, the mixed smooth Besov space is much wider class of mixed smooth Holder/Sobolev space. Roughly speaking, the mixed smooth Besov space consists of functions having form g(f_1(x_1),...,f_d(x_d)) where each f_i(x_i) is a function in a Besov space on [0,1] and g:R^d \to R is a sufficiently smooth function. Then, we can see that the m-Besov space includes an additive model \sum_{i=1}^d f_i(x_i) and a tensor model \sum_r \prod_{i=1}^d f_{r,i}(x_i) as special cases.
> We can also define an intermediate function class between the ordinary Besov space and the m-Besov space by taking a tensor product of B_{p,q}^s([0,1]^{d_1}), ..., B_{p,q}^s([0,1]^{d_K}) where d_1 + d_2 + ... d_K = d (if each d_i = 1, then it is reduced to the m-Besov space). We can also show a convergence rate which is between those of the m-Besov space and the Besov space, but we don't pursue this direction due to space limitation.

---

> ### Author Response · Authors · 2018-11-14
> **Reply from author (2/2)**
>
> (3)
> Q: Minor: - Defn of Holder class: you can make this hold for integral beta if you define m to be the smallest integer less than beta (e.g. beta=7, m=6). Imo, this is standard in most texts I have seen.
>
> A:
> Thank you for your detailed comment. Yes, it is one of very popular definitions of the Holder class for integer beta. On the other hand, one could also define it just by max_{|\alpha|<=beta}||D^\alpha f||_\infty where the last term is not involved (in other words, beta=m). Moreover, it could be defined by B_{\infty,\infty}^m. Unfortunately, for an integer \beta, these spaces do "not" coincide with each other. To avoid this kind of confusions, we decided to define the Holder space only for non-integer beta.
>
> (4)
> Q: - The authors claim that the approximation error does not depend on the dimensionality needs clarification, since N clearly depends on the dimension. ... what the authors meant was that the   exponential dependence on d has now been eliminated.
>
> A:
> Yes, the convergence rate is dependent on d. What we have meant by "exponential dependence on d is avoided" is that the dimensionality d is not coming directly to the polynomial order of n, that is, the exponent of the term n^{-2s/(2s + 1)}. Indeed, d also comes into the exponent of the log(n)-term as log(n)^d. However, comparing the polynomial order and poly-log order, poly-log order is milder. Then, we said "the curse of dimensionality is eased."
>
> (5)
> Q: Other - On page 4, what does the curly arrow notation mean?
>
> A:
> It means a continuous embedding. Namely, if X \hookrightarrow Y for two norm spaces X and Y, then X can be continuously embedded in Y (i.e., X is a subset of Y and there exists a constant C such that |x|_Y <= C |x|_X for x \in X). We have added the definition in the revised version.
>
> (6)
> Q:- Given the technical nature of the paper, the authors have done a good job with the presentation. However, in some places the discussion is very equation driven. For e.g. in the 2nd half of page 4, it might help to explain many of the quantities presented in plain words.
>
> A:
> We have added some text explanations in page 4. Due to space limitation, we could not give full expositions. But, we also added some explanations for the meaning of the approximation error rate and its relation to the depth, width and sparsity after Proposition 1 and Theorem 1.

---

### Public Comment · (anonymous) · 2018-11-07
**Your bound has curse of sample size!**

I am looking into the estimation error bound in Table 2 on Page 3.

We assume that \beta = 3, u = 0.1, and the sample size is large. Let's say n~exp(d).

Then we can reduce the bound to O(exp(-6d/7) * d^{0.88 d}).

The bound will blow up for large d.

Could you please clarify your results?

---

> ### Author Response · Authors · 2018-11-14
> **Dimensionality d is assumed to be constant (Reply from authors)**
>
> Thank you for your instructive question.
> First, in our analysis, the dimensionality d is a fixed constant and is not allowed to increase to infinity as the sample size n goes up. Thus, the curse of sample size does not occur for fixed d.
>
> Second, behind the order notation, there is a term depending on d. Actually, the log(n)^d term is originally comes from (log(n)/d)^d term (more precisely, it comes from D_{K,d} defined in Sec.3.2 where K will be O(log(n))). Thus, this term slowly increases (O(n^\epsilon) for a small constant \epsilon) under an assumption that d <= C log(n) for a sufficiently small C. On the other hand, for the convergence rate n^{-2s/(2s + d)} on the Besov space, d is not allowed to be log(n)-order. Actually, as long as d is log(n)-order, n^{-2s/(2s + d)} does not converges to 0. This contrasts the difference of the two settings, Besov and m-Besov settings.
>
> Finally, we also would like to remark that if d is O(log(n)), then the overall convergence rate will be changed. It will depend on the coefficient hidden in the order notation of d = O(log(n)). Showing the precise bound under this condition is out of paper's scope. Thus, we would like to leave that for the future work.

---

### Author Response · Authors · 2018-11-14
**Revision has been uploaded**

Thank you for your careful reading. We have uploaded a revised version.
The main difference from the original one is as follows:

1. Some additional text explanations are added for the definition of m-Besov space.
2. We added a few remarks for the approximation error bound in Proposition 1 and Theorem 1.
3. We have fixed some grammatical errors and typos.

Sincerely yours,
Authors.

---

### Meta-Review · Area_Chair1 · 2018-12-16
**Approximation of Besov spaces by Deep ReLU neural networks.**

**Confidence:** 4
**Recommendation:** Accept (Poster)

**Metareview:**

The paper extends the results in Yarotsky (2017) from Sobolev spaces to Besov spaces, stating that once the target function lies in certain Besov spaces, there exists some deep neural networks with ReLU activation that approximate the target in the minimax optimal rates. Such adaptive networks can be found by empirical risk minimization, which however is not yet known to be found by SGDs etc. This gap is the key weakness of applying approximation theory to the study of constructive deep neural networks of certain approximation spaces, which lacks algorithmic guarantees. The gap is hoped to be filled in future studies.

Despite the incompleteness of approximation theory, this paper is still a good solid work. Based on fact that the majority of reviewers suggest accept (6,8,6), with some concerns on the clarity, the paper is proposed as probable accept.